# T-cells produce acidic niches in lymph nodes to suppress their own effector functions

Hao Wu[1,2,7], Veronica Estrella[1,7], Matthew Beatty[3], Dominique Abrahams[1], Asmaa El-Kenawi[1,3], Shonagh Russell[1], Arig Ibrahim-Hashim[1], Dario Livio Longo[4], Yana K. Reshetnyak[5], Anna Moshnikova[5], Oleg A. Andreev[5], Kimberly Luddy[1], Mehdi Damaghi[1], Krithika Kodumudi[3], Smitha R. Pillai[1], Pedro Enriquez-Navas[1], Shari Pilon-Thomas[3], Pawel Swietach[6,8✉] & Robert J. Gillies[1,8✉]

The acidic pH of tumors profoundly inhibits effector functions of activated CD8 + T-cells. We hypothesize that this is a physiological process in immune regulation, and that it occurs within lymph nodes (LNs), which are likely acidic because of low convective flow and high glucose metabolism. Here we show by in vivo fluorescence and MR imaging, that LN para-cortical zones are profoundly acidic. These acidic niches are absent in athymic Nu/Nu and lymphodepleted mice, implicating T-cells in the acidifying process. T-cell glycolysis is inhibited at the low pH observed in LNs. We show that this is due to acid inhibition of monocarboxylate transporters (MCTs), resulting in a negative feedback on glycolytic rate. Importantly, we demonstrate that this acid pH does not hinder initial activation of naïve T-cells by dendritic cells. Thus, we describe an acidic niche within the immune system, and demonstrate its physiological role in regulating T-cell activation.

---

[1] Department of Cancer Physiology, H. Lee Moffitt Cancer Center and Research Institute, Tampa, FL 33612, USA. [2] Cancer Institute, Second Affiliated Hospital, Zhejiang University School of Medicine, 310058 Hangzhou, P.R. China. [3] Department of Immunology, H. Lee Moffitt Cancer Center and Research Institute, Tampa, FL 33612, USA. [4] Institute of Biostructures and Bioimaging (IBB), National Research Council of Italy (CNR), Turin, Italy. [5] Department of Physics, University of Rhode Island, Kingston, RI 02881, USA. [6] Department of Physiology, Anatomy and Genetics, University of Oxford, Parks Road, Oxford OX1 3PT England, UK. [7] These authors contributed equally: Hao Wu, Veronica Estrella. [8] These authors jointly supervised this work: Pawel Swietach, Robert J. Gillies. ✉email: pawel.swietach@dpag.ox.ac.uk; robert.gillies@moffitt.org

ymph nodes (LNs) are anatomically and physiologically complex organs that receive inputs from both lymphatic and blood vasculatures, and consist of discrete zones for processing and activating T and B cells (Fig. 1a). Despite their well-characterised histology and recent insights into the functional interplay between various resident cell-types and epithelial structures[1,2], relatively little is known of the physiological microenvironment of LNs in situ, and how it may influence immune cell functions. Notably, acidosis is known to inhibit effector T-cell functions under cell culture conditions and in solid tumours in vivo[3,4], but the relevance of this observation in the context of LN physiology has not been determined. Oxygen

**Fig. 1 Extracellular spaces of lymph node paracortical zones are acidic. a** Cartoon of lymph node (LN) showing zones occupied by T and B cells, blood vessels (B.V.) and the medulla (Me). Histological section of inguinal LN showing T-cell marker CD3 in paracortical zone ($N = 3$). **b** B6 mouse injected with pHLIP (40 μM in 60 μl) into footpad, followed by intravital imaging of inguinal LN 24 h later in window chamber. Left: composite image collected with ×1.6 objective for pHLIP-Cy5.5 (red; excited at 633 nm), autofluorescence (green; excited at 514 nm) and vasculature (blue; determined from transmission images). Right: montage of pHLIP fluorescence collected in overlapping fields of view with ×10 objective, summed across the depth of the LN ($n = 10$ mice). Experiment repeated on B6 mouse injected via the intraperitoneal cavity with **c** 200 μl of 12.5 mg/kg omeprazole (OME) and 1 mg/kg of bafilomycin (BAF) ($n = 4$). or **d** 200 μl of 5.2 mg/kg (5-N,N-dimethyl)amiloride (DMA) and 3.9 mg/kg acetazolamide (ATZ) 24 h prior to imaging ($n = 3$). **e** Experiment performed on athymic nude mouse, with same imaging settings, showing absence of pHLIP signal ($n = 8$). **f** Summary data for mean pHLIP fluorescence within LN boundary. Significance tested by one-way ANOVA with multiple comparisons ($N = 5, 10, 4, 4, 8, 11$); two sided at 5% significance. $p$-values compared to control: Depleted: $P = 0.0237$, Nude: $P = 0.0004$. **g** Intravital imaging of pH-sensitive cSNARF1 fluorescence in inguinal LN. Mice were injected with 70 kDa dextran-conjugated cSNARF1 into the tail-vein (20 mg/ml in 100 μl). Measurements on control mice, or mice treated with LPS ($n = 4$). **h** Statistical distribution of pHe data analyzed by Gaussian mixed models to separate pixels into clusters, representing compartments. Plots shows the pH-distribution in each of the LN compartments, averaged for all LNs. Note that compared to footpad injections, tail-vein injections detect an additional compartment corresponding to blood vessels. **i** Summary data for each LN compartment from 4, 3, 4, 3 LNs, respectively. **j** MRI-CEST pH imaging of control (B6) mice injected via i.v. with a 300 μl bolus of Isovue 370. pHe maps in inguinal LN region-of-interest are overlaid on anatomical $T_2$-weighted images. Mean ± SEM pHe measured in B6 ($n = 6$) and BALB/c ($n = 5$) mice. **k** Intravital imaging for hypoxic regions using 12.5 nmoles of ImageIT-Green hypoxic probe injected into B6 mice via the footpad in a 50 μl volume. As a positive control, LNs were made anoxic by bubbling PBS with $N_2$ and including the $O_2$-scavenger dithionite (1 mM), followed by cessation of circulation by cervical dislocation. Upper panels: composite image collected with ×1.6 objective for ImageIT-Green (green; excited at 514 nm) and vasculature (blue; determined from transmission images). Bottom panels: montage of ImageIT-Green fluorescence collected in overlapping fields of view with ×10 objective, summed across the depth of the LN ($N = 10$ control and 10 anoxia). (Scale bars = 1.0 mm for **b**–**e**, **k**; 0.5 mm for **g**).

tension in LNs has been reported to be low[5], and since hypoxic tissues are generally acidic via increased glucose fermentation, we hypothesized that LNs are also acidic.

Given the exquisite pH-sensitivity of cytokine release[3,4,6,7], LN acidity may have physiological consequences. For example, it may be advantageous to refrain from secreting inflammatory cytokines into a confined space of the LN. Aberrant activation of densely packed T-cells can, for example, induce immunopathological responses in both lymphoid and nonlymphoid tissues and, for that reason, many checkpoints are in place to prevent overactive lymphocytes in these organs[8–10]. While these checkpoints are active under physiological conditions, a pathologically overactive immune response can negatively impact lymph node structure and function. Persistent immune activation within lymphoid tissue, as seen with human immunodeficiency virus (HIV), results in lymph node fibrosis, often restricted to the T-cell zone, leading to diminished lymph node function and reduced peripheral T-cell numbers[11–13]. Additionally, high levels of cytokines accumulating within the T-cell zone would have detrimental effects on the acquisition of adaptive immunity. For example, IFNγ, whose expression is potently inhibited at low pH, alters T-cell polarization and homeostasis, can induce apoptosis, and inhibit lymphangiogenesis[14–16]. However, without direct measurements of pH in intact LNs, the physiological significance of this postulated regulatory influence is untested.

Here, we use in vivo fluorescence and magnetic resonance imaging to identify acidic niches in LNs. We further show that the source of this acidity is the T-cells themselves, based on measurements of lactic acid release and intracellular pH (pHi) in vitro, and the lack of acidity in LNs from athymic nude or lymphodepleted mice. We interpret the mechanism of LN acidity in terms of a steady-state between activated acid production and inhibitory feedback on glycolysis. We further show that the low extracellular pH (pHe) of LNs does not impair the ability of T-cells to become activated by antigen-presenting cells (APCs), whereas it does suppress elaboration of cytokine production. Our findings identify localized acidosis as a critical component of the adaptive immune response.

## Results

**Paracortical zones are acidic niches inside the lymph node.** The pHe of inguinal LNs in C57BL/6 (B6) mice was probed using

pH-Low Insertion Peptide (pHLIP) a short peptide that undergoes a conformational change at low pH to make it membrane penetrant, where it can be persistent[17–21]. To visualize areas of pHLIP insertion[20], the peptide was conjugated to the fluorophore Cy5.5, which emits in the far-red range for optimal signal-to-background ratio. To deliver the construct to the LN, injections (50 μl of 40 μM solution in PBS) were made into the right footpad (Fig. S1). After either 4 h or 24 h, the mouse was put under anaesthesia and its right inguinal LN was surgically exposed for intravital imaging in a window chamber, which was then mounted on an inverted laser-scanning confocal microscope[22]. Fluorescence and transmission images were taken with excitation lasers alternating between 633 and 514 nm, using either a low (×1.6) or higher power (×10) objective. Fluorescence (650–700 nm) excited at 633 nm revealed the distribution of pHLIP, which accumulates in acidic niches. High-power excitation at 514 nm evoked autofluorescence (550–650 nm), which was used to delineate the LN outline (hence size). Once optimized, the same imaging settings were applied consistently in all experiments using pHLIP. The ratio of transmission images acquired alternately at the two excitation wavelengths generated a ratiometric map that identified the vasculature on the basis of haemoglobin absorbance properties (greater absorbance at 514 nm, compared to 633 nm). Images obtained at various depths through an open pinhole and the 10x objective were summed to generate a projection of total fluorescence across the z-axis of the LN (Fig. S2a–c). To cover the entire area of the LN, high-resolution imaging was performed in overlapping fields of view, and the montage was assembled offline (MATLAB Control Point Selection tool). Analysis of pHLIP fluorescence indicated acidity in T-cell rich (CD3+) paracortical zones, with the notable absence of signal in B-zones in outer regions of the cortex (Fig. 1b, Fig. S2a). These data were further quantified in terms of the frequency-distribution of fluorescence intensity in the LN (Fig. S2d). This analysis indicated no difference between animals injected with pHLIP at 4 h or 24 h prior to imaging.

It is plausible that the source of acidity is an $H^+$-ion transport process of LN tubular structures, such as high endothelial venules[23,24]. Transporters underpinning such fluxes include omeprazole (OME)-sensitive P-type $H^+/K^+$-ATPases, bafilomycin (BAF)-sensitive V-type $H^+$-ATPases or 5-(N,N-dimethyl) amiloride (DMA)-sensitive $Na^+/H^+$ exchangers. Alternatively,

acidity may be attributable to the catalytic activity of membrane-tethered, acetazolamide (ATZ)-sensitive carbonic anhydrases. To test for the involvement of these acid-handling proteins, mice received intraperitoneal injections of pairs of inhibitors (OME/ BAF or DMA/ATZ) and a footpad injection of pHLIP, 24 h prior to imaging. These pharmacological interventions did not, however, abolish acidic niches in LNs arguing against the involvement of their target-proteins in acidifying the microenvironment (Fig. 1c/d, Fig. S2b). An alternative source of acidity may relate to the metabolic activity of T-cells residing in paracortical zones. Indeed, activated T-cells are known to have a substantial capacity to acidify media in vitro because of their high glycolytic fluxes[3,4,7,25,26]. To test this mechanism, pHLIP was injected into athymic nude mice that lack T-cells. In these animals, pHLIP no longer accumulated in paracortical zones (Fig. 1e, Fig. S2c), and its mean fluorescence decreased by ~80% (Fig. 1f). Because the LNs of nude mice may become altered by long-term effects of T-cell deficiency, we also tested whether acute lymphodepletion would result in decreased LN acidity. Figure S3 shows that injection of anti-CD4 and anti-CD8 antibodies successfully depleted 80% of CD3+ cells in the spleen and inguinal LN (Table S1). As shown in Fig. S2d and analysed in Fig. 1f, the LNs of acutely lymphodepleted mice had a significant, >50% decrease in pHLIP labelling. A statistical analysis of LN-averaged pHLIP fluorescence shows that LN pH is related to the number of T-cells residing in the LN (Fig. 1f).

While the spatial resolution of pHLIP fluorescence is excellent, acquired images lack the quantitative power to define the level of pHe in the LN. To determine this, B6 mice were injected with the membrane-impermeable 70kDa-dextran derivative of cSNARF1 (cSNARF1-Dex), a ratiometric dye that provides a calibratable readout of pH[27]. The settings used for imaging (×10 objective, excitation 514 nm, emission measured simultaneously at 580 ± 20 nm and 640 ± 20 nm) were optimised to minimise artefacts due to autofluorescence (Fig. S4a). A solution of cSNARF1-Dex dye in PBS (100 μL of 20 mg/mL) was injected into either the tail-vein or in the footpad, and fluorescence was subsequently measured by intravital imaging in anesthetized mice using a window chamber similar to that used for pHLIP imaging (Fig. S4b, c). These cSNARF1-Dex images were ratioed offline and converted to pHe maps using a calibration curve determined in vitro in buffered saline (Fig. S4d). High-quality images could be acquired 1 h after a tail-vein injection or 3 h after footpad injection. Specifically, tail-vein delivery of cSNARF1-Dex allowed for concurrent measurements of the pH inside blood vessels, which conveniently served as a reference for alkaline pHe. Analysis of pHe maps showed distinct areas of profound acidity in paracortical areas of the LN (Fig. 1g). The intensity histograms of pHe within the LN boundary were analysed by mixed Gaussian modelling to determine the number of compartments that best described the observed distribution (Fig. 1h). Notably, footpad injections produced a two-compartment distribution, whereas in tail-vein injections an additional alkaline compartment was observed, which was attributable to blood vessels. Irrespective of injection protocol, the most acidic LN compartments had a mean pHe of ~6.3, and were surrounded by regions of mean pHe ~6.7 (Fig. 1i). The pH inside blood vessels was ~7.1, as expected from venous blood draining from an acidic organ. To test if these acidic niches could become 'diluted' in an enlarged LN, or chemically neutralized with buffers, experiments were performed on mice injected with lipopolysaccharide (LPS; 100 ng/kg i.p. 48 h prior to imaging) to induce inflammation; or receiving oral bicarbonate (200 mmol/L of NaHCO₃ ad libitum 10 days before imaging), which has been shown to neutralize the acidic pH of tumors[28]. LPS treatment resulted in a significant (50%) increase in LN volume (Fig. S5), but no effect on pHe (Fig. 1i). There was

also was no effect of NaHCO₃ on pHe, arguing that the acidic niches in LNs are robustly regulated to a specific acidic level by a means of a feedback ("pH-stat") mechanism, which could not be disrupted by organ enlargement or systemic base-loading. Consistent with this hypothesis, the LNs of NaHCO₃-treated mice contained ~50% higher concentrations of lactate (Fig. S6a, b), despite no effect on pHe (Fig. 1i). In this scenario, the raised buffering power allows a greater cumulative glycolytic flux, reported in terms of lactate build-up because the magnitude of the negative feedback via pHe is lessened (Fig. S6a, b). To confirm that T-cells are the source of lactate, experiments on nude mice LNs showed significantly lower [lactate] compared to untreated control mice (Fig. S6c).

Since intravital imaging may potentially introduce artefacts from the necessary surgery, confirmation of low pHe in the LN was sought using a noninvasive method based on chemical exchange saturation transfer (CEST) magnetic resonance imaging imaging[29,30]. With this technique, images (albeit with inferior resolution to fluorescence microscopy) are collected from different saturation frequencies in mice injected with the CT contrast agent iopamidol (Isovue; Fig. S7a). This compound has ionizable secondary amides that resonate at different frequencies (Fig. S7b), which can be interrogated using frequency-specific excitations. Since these resonances have distinct pH profiles (Fig. S7c), the ratios of saturation occurring at the two frequencies report pHe (Fig. S7d). The inguinal LNs of B6 and BALB/c mice were determined to have a mean pH of ~6.4 (Fig. 1j), which is consistent with measurements obtained by cSNARF1 fluorescence microscopy.

Previous reports have suggested the existence of hypoxic regions within the LN using flow cytometry of LN derived cells exposed to the hypoxia adduct, pimonidazole[5]. Depletion of oxygen could influence T-cell functions, either by acting alongside the influence of low pH, or as the dominant modulator, with low pHe merely being a collateral epiphenomenon[31]. To seek evidence for hypoxic niches and their relationship with pHe, B6 mice were injected intraperitoneally with the hypoxic probe pimonidazole; and 1 h later, inguinal LNs were excised and fixed for immunohistochemistry. The pattern of pimonidazole staining was sparse and weak, which argued against substantial hypoxia (Fig. S8a). As a further test, the fluorescent hypoxia probe ImageIT-Green was injected into the footpad of B6 mice and, after 4 h, the inguinal LN was surgically exposed for intravital imaging (excitation at 488 nm, emission 540 ± 20 nm) in a window chamber. ImageIT-Green fluorescence is irreversibly increased in regions with O₂ tension lower than 5%, and thus provides an independent assessment of hypoxic niches, even when the LN is surgically exposed to the atmosphere. LNs emitted only low levels of autofluorescence above background (Fig. S8b), and similarly low levels of fluorescence following injection of ImageIT, indicating that oxygen tension in the LN is normally greater than 5% (Fig. 1k). As a positive control, LN anoxia was produced by bathing the organ in deoxygenated (N₂-bubbled) PBS that contained the oxygen-scavenger, dithionite (1 mM), followed by cessation of circulation by means of cervical dislocation. Under these conditions, ImageIT fluorescence (and hence hypoxia) was abundant (Fig. 1k). Thus, LN paracortical zones are profoundly acidic, but not substantially hypoxic.

**Acidic niches result from activated T-cells lactic acid.** When activated in vitro, T-cells undergo a dramatic increase in aerobic glycolysis, considered necessary for engaging effector T-cell functions[3,4,7,25,26]. Activation of T-cells with plate-bounded anti-CD3ε antibody and soluble anti-CD28 antibody had a rapid and robust effect on the extracellular acidification rate

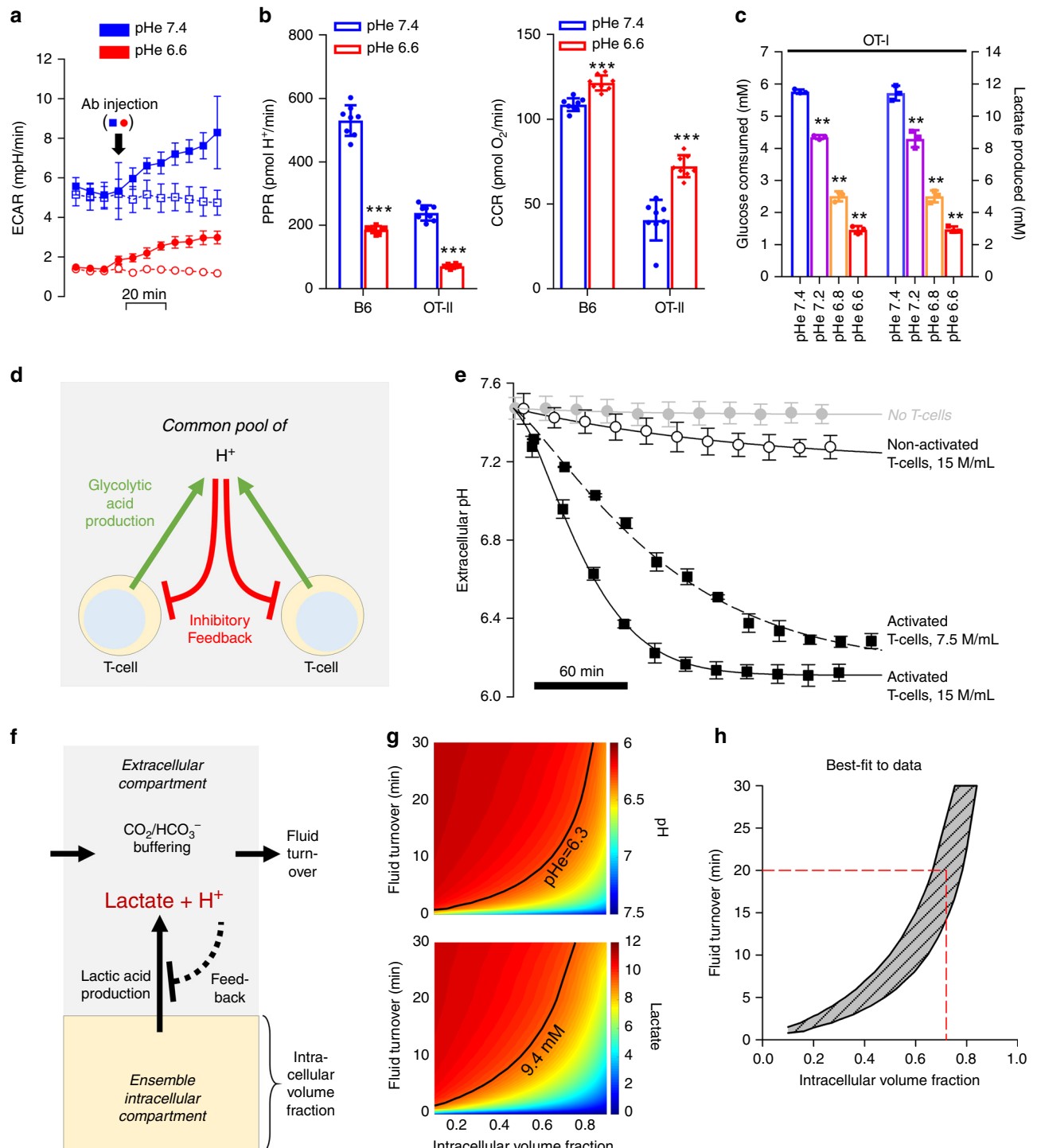

(ECAR), measured by a Seahorse extracellular flux (XF) analyser (Fig. 2a). Note that although ECAR was lower in T-cells incubated at a lower pH, addition of CD28 Ab increased the glycolytic ECAR at both low and high pH. ECAR can be converted, using data for buffering capacity, to a quantitative $H^+$ production rate (PPR) generated by T-cells, 48 h after their activation (Fig. 2b). At pHe 7.4, the PPR of activated T-cells, isolated from B6 mice, was equivalent to ~10 mmoles/min/(L of intracellular volume), after accounting for buffering capacity ($\beta = 3.78$ mM/pH) and intra/extracellular volume-ratios (chamber volume of 2.28 $\mu$L, cell radius cell radius = 5.14 $\mu$m, SD = 0.68 $\mu$m, number of cells 100,000). This glycolytic rate is high; comparable to the most

metabolically-active cancer cells[32–37], and may underpin the low pHe observed in vivo in paracortical zones (Fig. 1a–j), provided that the flux is sufficiently large in relation to fluid clearance by perfusion of the LN. Steady-state lactate levels in inguinal LNs of B6 mice were $9.4 \pm 3.5$ mM ($N = 8$), which is higher than levels in LNs of nude mice ($2.1 \pm 0.9$ mM; $N = 8$; Fig. S6c). Assuming that the fluid clearance is comparable, these data indicate that the products of T-cell glycolysis accumulate in LNs and are not rapidly washed away with perfusion.

When metabolic flux measurements were repeated in media at a reduced pHe, glycolytic flux decreased profoundly, along with a concomitant increase in the $O_2$ consumption rate, OCR (Fig. 2b).

**Fig. 2 Feedback regulation of T-cell glycolysis by pH establishes an acidic extracellular milieu at the steady-state. a** Time course of extracellular acidification rate (ECAR) was measured by Seahorse in B6 T-cells (Mean ± SD, $n = 7$ biological samples.). Injection of activating antibody (or vehicle for control) at 20 min evoked an increase in ECAR, due to the activation of T-cell glycolysis. **b** Proton production rate (PPR) measured by Seahorse in B6 or OT-II T-cells is reduced under acidic conditions. In paired experiments on B6 or OT-II T-cells, oxygen consumption rate (OCR), measured by Seahorse, is increased under acidic conditions (two-tailed, unpaired $t$-test, mean ± SD, $n = 8$ biological samples. PPR (B6, $p = 1.35E-11$; OT-II, $p = 1.62E-11$), OCR (B6, $p = 2.22E-5$; OT-II, $p = 1.14E-5$). Asterisks (***) represent $p < 0.0001$. **c** Glucose consumption and lactate production as a function of pHe in OT-I T-cells, expressed as mean ± SD; $n = 3$ biological samples. Significance tested by one-way ANOVA with multiple comparisons $p < 0.001$. **d** Schematic diagram of feedback loop between lactic acid production by glycolysis, and its inhibitory feedback by extracellular pH. **e** Time course of pHe measured in 60 μl volumes of 5 mM HEPES-buffered media containing no cells, nonactivated T-cells or activated (CD3-coated plates, then incubated in media containing 2 μg/ml CD28) T-cells at the densities indicated. $n = 3$ biological replicates. Data shown as mean ± S.E.M. Activated T-cells acidify the restricted extracellular volume towards pH 6.3 within several hours. **f** Schematic representation of mathematical model used to simulate the relationship between extracellular pH and lactate for a system featuring glycolytic lactic acid production and feedback inhibition by extracellular pH, as determined from panel **e** (i.e. linear inhibition towards zero production at pH 6.3), for a LN paracortex of intracellular volume fraction $v_i$, and fluid turnover (perfusion) of $\tau$. (**g**) Results of simulation for extracellular pH (upper panel) and lactate (lower panel). Black line shows the combination of $v_i$ and $\tau$ that simulates experimentally observed data for pHe (6.3; Fig. 1) and lactate (9.4 mM; Fig. S5). **h** Replotting of the best-fit curves from panel **g**. Red dashed line shows solution of this mathematical problem using the literature value for $\tau$ of 20 min. This indicates that ~70% of the paracortical zone is occupied by T-cells, engaged in lactic acid production, the source of low pHe measured in LNs.

Although the increase in OCR (and hence oxidative ATP production) was modest, it is sufficient to compensate for a loss of glycolytic ATP production, assuming a ratio of ca. 18:1 (Fig. 2b). These results were also observed with OT-II (CD4+) T-cells stimulated with OVA$_{323-339}$ (ISQAVHAAHAEINEAGR) peptide (Fig. 2b). These observations would argue that once pHe had attained a low level, further acidification would be curtailed, leading to steady-state. Since Seahorse measurements of rate are performed over short periods of time, they cannot determine if the high glycolytic rate of activated T-cells could be sustained long enough to produce a meaningful accumulation of lactic acid in vivo. To test this, OT-I (CD8+) T-cells, stimulated with OVA$_{257-264}$ (SIINAFEKL) peptide, were plated at low density (500,000 cells/ml) at pH = 7.4. After 24 h, the media had accumulated ~10 mM lactate (Fig. 2c), confirming the cells' capacity to sustain a high glycolytic flux. Both lactate production and glucose consumption decreased with decreasing pHe (Fig. 2c), consistent with ambient acidity feeding back negatively on glycolysis.

The scheme shown in Fig. 2d illustrates the proposed relationship between extracellular acidity and T-cell glycolysis. Here, pHe is predicted to stabilize at an acidic level once the inhibitory feedback reaches a level that fully blocks any further acid production by glycolysis. To determine if steady-state pHe could be attained within a reasonable time-frame, T-cells from B6 mice were resuspended in lightly buffered media (2 mM Hepes + 2 mM Mes) and plated at either 7.5 or 15 million/mL in small (60 μL) inspection chambers. Media of low buffering capacity (~3.78 mM/pH over the range 6.0–7.5) were chosen for this experiment to allow metabolism from a relatively low density of cells to measurably affect pH. Dextran-conjugated cSNARF1 (0.25 μg/μl) was added to media to report extracellular acidification in real-time (Fig. 2e). In this in vitro system, nonactivated (naïve) cells represented LN-resident T-cells, and in separate experiments, T-cells were activated to enhance metabolic rate further. Both activated and naive T-cells caused a progressive decrease in pHe, until this reached an acidic steady-state of pHe = 6.3 within 2–10 h for activated and nonactivated T-cells, respectively. Notably, in vivo pH measurements using dextran-cSNARF-1 (Fig. 1g, h) also showed a pHe of ~6.3. The average density of T-cells in LNs of B6 mice is 800 million/mL (Fig. S9). At these densities, activated and naïve T -cells will, respectively, produce ~14 and 1.5 mM H+ per hour while in the inguinal LN, and even at the lower rate, a steady-state pHe of 6.3 is attainable within 16 h, assuming restricted capillary perfusion. Activation of T-cells thus serves to hasten the rate at which pHe stabilizes at its pH-

stat. In summary, our in vitro findings and in vivo correlates indicate that activated T-cells can profoundly and rapidly acidify their milieu, and maintain it at a reduced pHe.

To relate the in vitro findings to the conditions that prevail inside LNs, a mathematical model was used to simulate steady-state pHe and lactate levels. The volume of a mouse inguinal LN is typically 2.5 μL[38], half of which is occupied by the paracortex[39], and contains 1–4 million T-cells[40]. Only <10% of flow from the afferent vessel supplying LNs perfuses the central regions comprising the cortex and medulla, with the remaining flow takes a peripheral route[41]. Central flow is equivalent to 5% of LN volume per minute[41], suggesting that the turnover of fluid in the central region occurs in ~20 min, i.e. a relatively slow wash-out which would favour metabolite build-up. The mathematical model, presented schematically in Fig. 2f, includes variables describing glycolytic rate, its pHe-dependence, fractional volume of the intracellular space ($v_i$), and turnover of extracellular fluid ($\tau$). Briefly, the intracellular compartment releases lactic acid into a poorly-perfused extracellular space buffered with $CO_2/HCO_3^-$, where lactate and H+ can accumulate and inhibit, via pHe, the glycolytic rate. Results of simulations for various combinations of $v_i$ and $\tau$ are shown in Fig. 2g; the thick line shows those combination that predict experimentally measured values for pHe (6.3; Fig. 1g) and lactate (9.4 mM; Fig. S6c). Replotting these curves in Fig. 1h shows the range of $\tau$ and $v_i$ compatible with experimental data. Since $\tau$ is estimated[41] to be ~20 min, the best-fitting $v_i$ is predicted to be ~0.7, i.e. a combined T-cell volume of ~0.9 μL (70% of half the LN paracortex volume, 1.25 μL) that equates to 1.5 million T-cells. These values are well within the range of measurements in LNs, arguing that the degree of acidity measured in LNs can be adequately described by T-cell metabolism, as shown in the model Fig. 2d.

**Acid inhibits T-cell lactic acid efflux and glycolysis.** Glycolytic inhibition at low pHe was studied further in terms of its dynamics using the Seahorse analyser. Injection of a volume of HCl acid, determined a priori to reduce the pH of lightly (2 mM) HEPES/MES-buffered medium from 7.4 to 6.6, triggered a rapid fall in extracellular acidification rate (ECAR) in activated T-cells from B6 mice, which reversed with an injection of NaOH that restored pHe back to 7.4 (Fig. 3a). Thus, the effect of acidosis on glycolytic rate is acute and reversible, and its mechanism may involve a dynamic resetting of pHi, which was tested in T-cells loaded with cSNARF1, calibrated with nigericin (Fig. S10a) and imaged confocally. Changes in pHe, produced by switching between superfusates titrated to pH 7.4 or pH 6.6, evoked dynamic

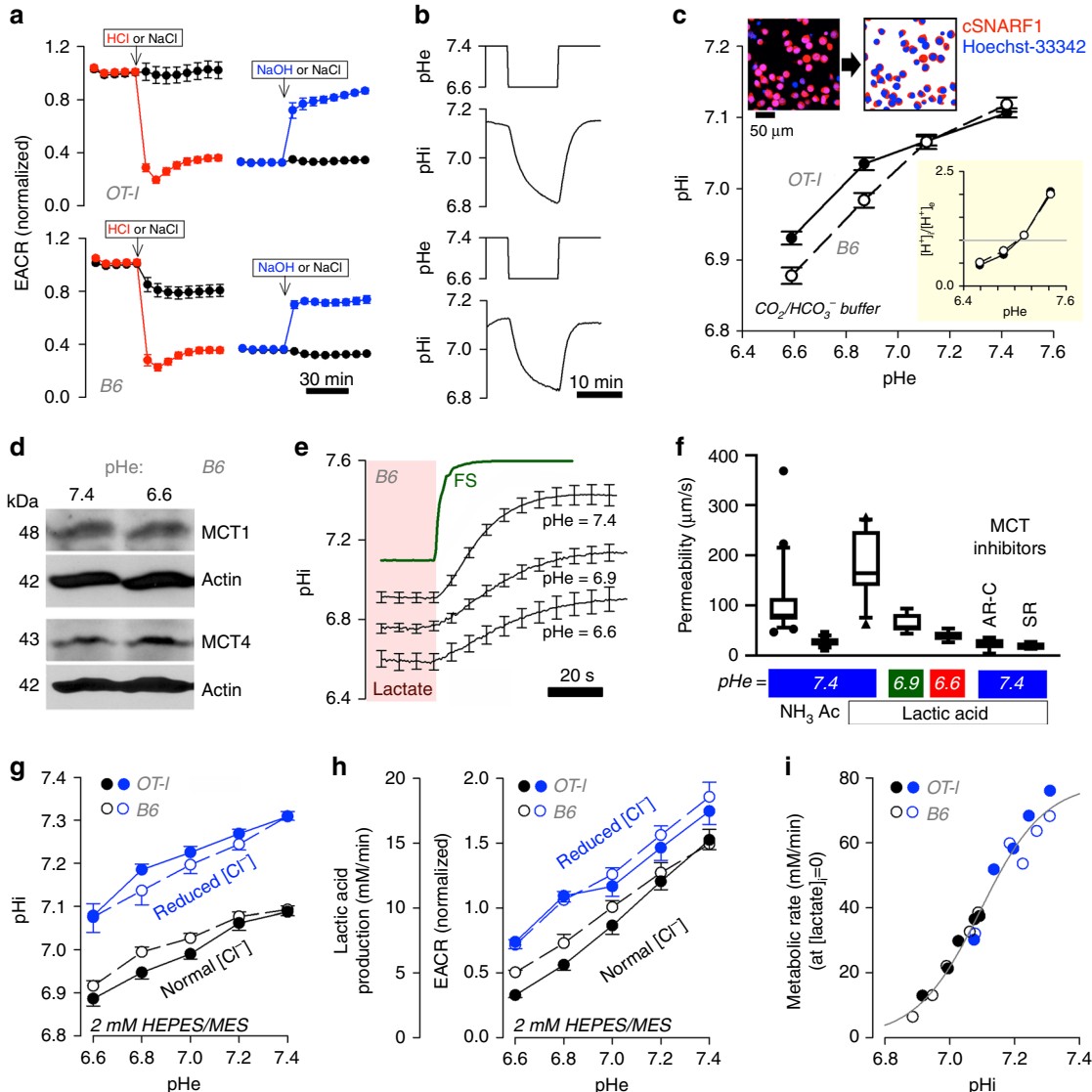

**Fig. 3 Mechanism of T-cell glycolysis inhibition by low pH. a** An injection of HCl abruptly reduces extracellular acidification rate (ECAR) in OT1 and B6 T-cells; this reverses upon an injection of NaOH. NaCl injections performed as sham controls. Solutions were lightly buffered with 2 mM HEPES/MES mixture and titrated to desired pH. Mean ± SEM ($n = 4$ biological replicates). **b** A reduction in extracellular pH (pHe) evokes a delayed fall in intracellular pH (pHi), as measured from cSNARF1 fluorescence (2 mM HEPES/MES mixture). Mean of 10 time course recordings; error bars not shown for clarity. **c** Fluorescence imaging of cells under superfusion with $CO_2/HCO_3^-$ buffer. Cells co-loaded with cSNARF1 (red) to report pH and Hoechst-33342 (blue) to exclude nuclear areas from the analysis. Plot shows relationship between pHe and pHi at the steady-state in OT1 and B6 cells. Note the transmembrane [H+] gradient, shown in inset, inverts near resting pHi. Mean ± SEM of 5 recordings of fields of view containing 40–60 cells. **d** Western blot for MCT1 (48 kDa) and MCT4 (43 kDa) relative to actin (42 kDa) on lysates collected from B6 T-cells that had been incubated at pHe 7.4 or 6.6 (N = 3). (See Supplementary Fig. S17 for full blot). **e** Measuring total MCT activity from the rate of pHi change driven by transmembrane lactate efflux. T-cells under superfusion were equilibrated with one of the three conditions, 30 mM lactate at pHe 7.4, 15 mM lactate at pHe 6.9 or 7.5 mM lactate at pHe 6.6. Note that, for lower pHe, the lactate concentration was reduced to ensure that comparable levels of lactic acid are present at equilibrium. Rapid switching to lactate-free solution at the same pHe evoked net lactate efflux. Apparent permeability to lactic acid can be calculated from the rate of pHi change, buffering capacity and transmembrane gradient. To confirm that the ensuing pHi response was not rate-limited by the speed of solution exchange, one solution was labelled with fluorescein sulphonic acid (FS) and the rate of fluorescence-change indicated an exchange time constant of 2.6 s. Mean ± SEM of 10 cells per condition. **f** Apparent membrane permeability for $NH_3$ (added as 15 mM $NH_4Cl$; $n = 21$) acetic acid (Ac; 30 mM NaAcetate; $n = 12$) and lactic acid (7.5–30 mM Na Lactate) at high ($n = 12$), intermediate ($n = 6$), and low ($n = 8$) pHe. Indicated experiments performed in the presence of MCT inhibitors AR-C (AR-C155858; 10 μM; $n = 7$) and SR (SR13800; 10 μM;; $n = 7$). Mean ± S.E.M. of 7–15 cells per condition. Box shows median and 25–75% percentiles and whiskers show 10–90% percentile. **g** Steady-state relationship between pHe and pHi mapped for 2 mM HEPES/MES solution containing either normal (140 mM) or reduced [Cl] (7 mM), iso-osmotically substituted with gluconate to offset pHi at constant pHe. Mean±SEM of 6 recordings with 40–60 cells each. **h** EACR, calibrated to units of lactic acid-production rate (mM/min), is shown not to be a unique function of pHe; Data shown are Mean ± S.D., $n = 14$ wells over two independently seeded plates. **i** Data from **g** and **h** analyzed to generate a relationship between metabolic rate, extrapolated to lactate-free conditions (see Eq. (1)). Best-fit is a simple function of pHi, described by a Hill curve.

changes in pHi in the same direction but with a short delay and of reduced amplitude (Fig. 3b). To determine if these pHi shifts were stable over a longer period of time, T-cells were equilibrated in media over a range of pHe, and their pHi was measured once it reached steady-state. Upon decreasing pHe from 7.4 to 6.6, pHi stably decreased by ~0.2 pH units (Fig. 3c). Critically, this transduction of a pHe change into a sustained pHi signal allows access to a myriad of protonatable targets in the cytoplasm[42], including enzymes in the glycolytic pathway[43], such as highly pH-sensitive phosphofructokinase-1 (Fig. S10b).

The coupling between pHe and pHi arises from changes in transmembrane acid-base traffic, including that carried by $H^+$-monocarboxylate co-transporters (MCTs). Low pHe thermo-dynamically hinders $H^+$-lactate export[44], leading to an intracellular retention of $H^+$ and lactate ions. Both MCT1 and MCT4 are present in T-cells, and their expression remains stable even at low pHe (Fig. 3d; S17). Flux carried aboard MCTs was quantified from the rate of pHi change evoked by the withdrawal of extracellular lactate, which was performed using a rapid perfusate switching system[44] (Fig. 3e). MCT transport capacity, quantified in terms of an apparent permeability to lactic acid, was 180 μm/s in T-cells from B6 mice (Fig. 3f). The measured permeability to acetic acid (a smaller organic acid) and $NH_3$ (a gas) were lower (~110 and ~30 μm/s, respectively), indicating that lactate is transported by means of a protein-facilitated process. This transporter is likely a MCT, as the inhibitors AR-C155858 and SR13800 (10 μM) reduced lactic acid permeability to 20 μm/s, i.e. to the level of protein-unassisted permeability across the lipid matrix (Fig. S10c, Fig. 3f). When measurements were repeated at lower pHe (6.6 and 6.9), lactic acid permeability was reduced substantially, consistent with a thermodynamic inhibition of MCT1. Reduced MCT transport capacity ultimately leads to an intracellular acidification and lactate retention, both of which feedback negatively on glycolysis. Given that glycolytic flux is ultimately limited by the rate of end-product removal, this action of pHe can explain the glycolytic suppression attained at low pHe, even in activated T-cells with high glycolytic capacity[4,45].

To confirm that changes in pHi mediate the pHe-glycolysis relationship, ECAR measurements were performed under conditions that selectively manipulated pHi at constant pHe. To raise pHi in acidic media (a 'rescue'), NaCl in the perfusate was reduced by iso-osmotic replacement with Na-gluconate (Fig. 3g). This ionic substitution alters the transmembrane $Cl^-$-driving force for acid-loaders (including $Cl^-/HCO_3^-$ exchangers), which leads to import of $HCO_3^-$ into cells. Strikingly, the glycolytic rate ($J_{glyco}$), reported as ECAR, was not a unique function of pHe (Fig. 3h); instead, it could be described by a simple mathematical function of pHi and intracellular [lactate]:

$$J_{glyco} = J_{glyco}^{MAX} \times \frac{K^n}{[H]i^n + K^n} \times \frac{Q}{[lactate]i + Q} \qquad (1)$$

Where K and Q are the apparent binding constants for $H^+$ and lactate ions, respectively, and n is the Hill coefficient for the binding of $H^+$ ions. Knowing the transmembrane pH gradient (Fig. 3c), metabolic rate (Fig. 2a), and membrane permeability to lactic acid (Fig. 3f), it is possible to predict cytoplasmic [lactate] at steady-state (Table S2) and, by best-fitting these data to Eq. (1), describe the relationship between pHi and glycolytic flux (Fig. 3i). This relationship, determined for both B6 and OT1 T-cells, was highly cooperative ($n = 4.39$) and with half-maximal activation near resting pHi ($-\log(K) = 7.095$), i.e. consistent with a highly pH-sensitive response. The effect of end-product inhibition by lactate ions was best described by Q of 2.1 mM. To test this model, some predictions of Eq. (1) were confirmed experimentally. At constant pHe and pHi, the addition of lactate is expected to reduce glycolysis by end-product inhibition. Since the L- and

D- isoforms are transport substrates for MCT, both will similarly influence transmembrane traffic, but only the L-isomer will produce end-product inhibition via stereo-specific lactate dehydrogenase (LDH). Indeed, the L-isoform produced a stronger inhibition of ECAR (Fig. S10d; to 21% of control vs. 58% of control with D-lactate). Consistent with this, Eq. (1) predicts that, at constant pHi, L- and D-lactate would respectively reduce $J_{glyco}$ to 24% and 44% of control.

**Acid suppresses cytokine release but not T-cell activation.** The powerful inhibition of glycolysis at low pHe is expected to suppress effector T-cell functions[3,4]. In keeping with previous findings[7,25,26], CD3/CD28-activated T-cells isolated from B6 mice had dramatically reduced interferon-gamma (IFNγ) secretion when incubated for 24 h at pHe 6.6, compared to time-matched controls incubated at pHe 7.4 (Fig. 4a). The reduction in measured IFNγ at low pH was not an artefact of a conformational disruption to the epitope detected by the ELISA method because immunoreactivity for known quantities of synthetic IFNγ remained stable over a wide range of pHe (Fig. S11a). Acid inhibition of IFNγ elaboration was titratable in four different strains of T-cells: (i) B6 T-cells stimulated with anti-CD3/anti-CD28, (ii) OT-I (CD8+) T-cells stimulated with OVA_{257–264} peptide, (iii) Pmel (CD8+) T-cells stimulated with gp-100 peptide, and (iv) OT-II (CD4+) T-cells stimulated with OVA_{323–339} peptide (Fig. 4b). The inhibitory effect of acid was reversible: T-cells stimulated at pHe 7.4 had reduced IFNγ secretion when restimulated at pH 6.6, but could readily resume IFNγ production when transferred back to an alkaline environment (Fig. 4c). The actions of acidity were not limited to IFNγ, as IL-2 release was also inhibited at low pH (Fig. 4d). Acid inhibition of cytokine release was not attributable merely to a generalized failure of exocytosis, as measurements on a large panel of cytokines indicated that, whilst most had reduced secretion at low pHe, some cytokines (MDC, MIG, and IP-10) showed greater secretion in acidic conditions (Fig. 4e; Fig. S11b, c). Although there was an inhibitory effect on cytokine secretion, acidosis did not consistently inhibit the proliferation rate of B6, Pmel, OT-I, or OT-II T-cells stimulated with anti-CD3 antibodies (Fig. 4f) or specific antigen (data provided with review).

While we have shown that a major effect of pHe on T-cell activation is mediated via inhibitions of glycolysis, it does not rule out specific acid-mediated ligand interactions; such as VISTA binding to co-inhibitor PSGL-1[46]. Hence, it was important to investigate other candidates for parallel mechanisms linking acidosis with reduced cytokine elaboration. Other potential $H^+$-sensing mechanisms include, inter alia, activation of acid-sensing receptors or channels[47] or the modulation of $Ca^{2+}$ signalling[48]. Two acid-sensing G-protein coupled receptors, GPR65 (TDAG8) and GPR68 (OGR1), are expressed in T-cells[7], but activated T-cells obtained from Gpr65- or Gpr68-knockout mice remained inhibited under acidic conditions (Fig. S12a). Furthermore, small-molecule inhibitors of OGR1 and GPR4 (BA-39-PQ30-1, NE-52-QQ57-1, gifts from Novartis) failed to rescue cytokine release under acidic conditions (Fig. S12b, c). Inhibition of TRPV1, an acid-sensing ion channel, did not rescue IFNγ production either (Fig. S12d). Acid-sensing ion channel (ASIC)[49,50] isoforms 1 and 3 are also expressed in T-cells[7], but the potent and specific inhibitors, A-317567, APETx2, and psalmotoxin[51,52], were unable to restore T-cell function at low pH (Fig. S12e–g). Similarly, amiloride and cariporide showed no 'rescue' effect on IFNγ (Fig. S12h, i). The only treatments found that at least partially raised IFNγ production at acidic pH were phorbol esters and histone deacetylase inhibitors (Fig. S12j/k), but these effects were only modest. T-cell $Ca^{2+}$ signalling is thought to respond to low

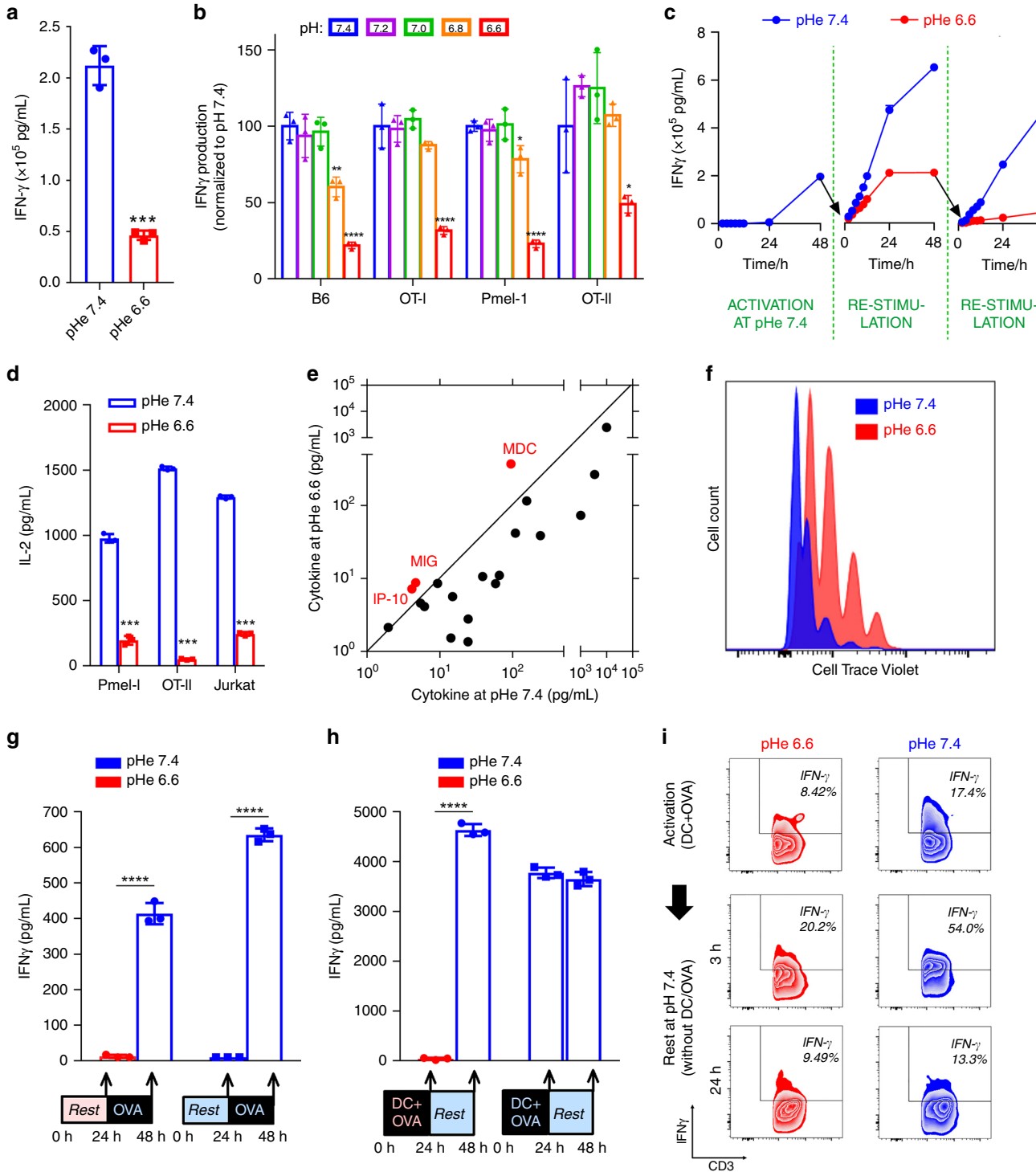

pHe through the pH-sensitivity of Ora1 Ca$^{2+}$ channels[53,54]. However, low pHe did not meaningfully change store-operated Ca$^{2+}$ entry interrogated by a standard protocol (Fig. S13). Thus, ruling out the contribution of these other mechanisms leads us to conclude that inhibition of glycolysis via the intracellular build-up of lactate and H$^{+}$ ions are responsible for T-cell inactivation by low pH.

Results thus far indicate that T-cells residing in restricted niches of paracortical zones will produce an acidic steady-state pHe as their lactic acid output comes into balance with feedback inhibition through reduced MCT activity and glycolytic flux (Fig. 3e). The attained steady-state pHe is sufficient to suppress

cytokine secretion (Fig. S12a–k), and while this would protect the LN from damage caused by cytokines, a comparable inhibitory effect on the ability for T-cells to undergo activation would be deleterious to the acquisition of adaptive immunity. To test if exposure to acidity also affects antigen activation, naïve T-cells isolated from OT-I mice were incubated, without stimulation, at either pHe 6.6 or 7.4 for 24 h. Cells were subsequently activated with OVA$_{257-264}$ peptide at pHe 7.4. Measurements of IFNγ secretion performed 24 h later showed significant IFNγ secretion (measured by ELISA) in both groups, indicating that preconditioning at low pH did not impair the ability of T-cells to be activated (Fig. 4g). The extent of T-cell activation was also

**Fig. 4 T-cell effector functions are inhibited at acidic pH. a** Interferon γ (IFNγ) production from C57BL/6 (B6) T-cells is reduced at low pHe, as determined by ELISA; $n = 3$, $p = 0.00013$. **b** INFγ production, measured over a range of pHe in T-cells from B6 mice as well as three antigen-specific strains. $n = 3$. IFNγ levels were compared with those at pHe 7.4 within each strain. B6 (pHe 6.8, $p = 0.0034$; pHe 6.6, $p = 0.00013$), OT-I (pHe 6.6, $p = 0.0013$), Pmel-1 (pHe 6.8, $p = 0.017$; pHe 6.6, $p = 6.25E-6$), OT-II (pHe 6.6, $p = 0.046$). **c** Time course of IFNγ levels in media following pH-manoeuvres that demonstrate the reversal of acid inhibition upon subsequent exposure to alkaline pH (rescue experiment); $n = 3$. **d** Interleukin-2 (IL-2) release, measured by ELISA in Pmel-1 and OT-II T-cells and a Jurkat leukaemia cell line, is reduced at low pHe; $n = 3$. Pmel-1, $p = 8.77E-6$; OT-II, $p = 6.85E-9$; Jurkat, $p = 5.64E-8$. **e** Relationship between cytokine levels at low and high pH, determined in paired experiments by the Cytokine Beads Array (CBA) assay. For most cytokines, with the exception of those highlighted in red (IP-10, MIG, MDC), acidic conditions evoked a reduction in release. **f** Rate of B6 cell proliferation measured by CellTrace Violet assay. **g** IFNγ production was measured, by ELISA, at the end of a 24 h preconditioning period (no OVA added) at either pHe 6.6 or 7.4, and then at the end of a consecutive 24 h period in the presence of antigen (OVA) at pHe 7.4. IFNγ production can be activated irrespective of whether cells had been preconditioned at pHe 6.6 or 7.4; $n = 3$. pHe 6.6 precondition, $p = 2.82E-5$; pHe 7.4 precondition, $p = 4.25E-7$. Asterisks (****) represent $p < 0.0001$. **h** IFNγ production by T-cells activated with dendritic cells (DC) and antigen (OVA) measured after 24 h at pHe 6.6 or 7.4, followed by measurements at the end of a subsequent 24 h period without stimulation at pHe 7.4 (rest). T-cells can become activated by DC/OVA at acidic or alkaline pHe, and fully retain the capacity to produce cytokines when transferred to alkaline media; $n = 3$. pHe 6.6 activation, $p = 3.22E-7$; pHe 7.4 activation, $p = 0.29$. Asterisks (****) represent $p < 0.0001$. **i** Flow cytometry. Intracellular IFNγ staining of T-cells activated with DC and antigen (OVA) measured after 24 h of treatment in either pHe 6.6 or 7.4 (top panels). Cells were then transferred to pHe 7.4 to rest in the absence of DC and OVA, and measurements were performed after 3 h of resting. All the experiments were repeated at least twice and expressed as mean ± SD and analyzed by two-tailed, unpaired $t$-test unless indicated otherwise. Significance level: *$p < 0.05$; **$p < 0.01$; ***$p < 0.001$; ****$p < 0.0001$.

assayed in terms of the proportion of IFNγ positive (IFNγ$^+$) cells, determined by flow cytometry at 3 h and 24 h after activation with peptide. A significant increase in the percentage of IFNγ$^+$ cells was observed after 24 h in both the control and acid preconditioned groups (Fig. S14; S18).

To evaluate whether low pHe can affect T-cell activation by APCs, the process was modelled in vitro by co-culturing T-cells with monocyte-derived dendritic cells (DCs) and antigen (OVA$_{257–264}$) at either pHe 7.4 or 6.6. No differences in the expression of the DC marker CD40 were observed at either low or control pHe, thus any potential actions of acid cannot be argued in terms of insufficient stimulation by DCs (Fig. S15a). DCs efficiently took-up FITC-tagged OVA protein (Fig. S15b) and presented OVA$_{257–264}$ peptide equally avidly at pHe 6.6 and 7.4, indicating that acidosis did not impair the ability of DCs to process and present antigen to T-cells (Fig. S15c). These findings are in broad agreement with prior reports demonstrating that low pHe does not attenuate DC antigen-presenting activity[55,56]. T-cells activated by DCs at the reduced pHe of 6.6 produced less IFNγ at 24 h after primary activation, but a further 24-h period in alkaline conditions without the continued presence of DCs fully restored IFNγ production (Fig. 4h). Intriguingly, only a 3-h period of rest at pHe 7.4 was sufficient to increase the number of IFNγ-positive cells measured by flow cytometry (Fig. 4i; S18), indicating that the inhibitory actions exerted on T-cells by acidosis in the LN can be reversed once the T-cells re-enter the circulation. The restoration of effector functions, assayed in terms of IFNγ-positive cells, was also observed when T-cells were activated with OVA$_{257–264}$ peptide, irrespective of the presence of DCs (Fig. S16). In summary, T-cell effector functions could be restored promptly upon exposure to alkaline conditions, even if the activation by DCs had occurred at acidic pHe.

## Discussion

We describe a naturally occurring acidic niche in the body, one of a few sites wherein low pH may play an integral role in normal physiological function. Specifically, our results demonstrate a potential role for the LN microenvironment in shaping T-cell biology. Within the structurally-restricted extracellular spaces of paracortical zones, T-cells activated by antigen-presenting cells (e.g. DCs) produce an acidic environment, set by the balance between the enhanced capacity to generate lactic acid glycolytically and the ensuing negative feedback exercised by acid inhibition of MCT and glycolytic enzymes. Whilst this low pHe does not block the process of activation by antigen, it will suppress the

production and release of many (but not all) cytokines, thereby possibly protecting the LN from premature and unwarranted release of inflammatory and anti-inflammatory cytokines. The complexity of these cytokines' interactions within a LN are poorly understood and perhaps one function of this acid-induced inhibition of T-cells is just to simplify this milieu within the confined space of a LN. Once outside the acidic LN, effector functions of egressing T-cell become rapidly uninhibited. This effect of pH on T-cells is consistent with the emerging notion that "the role of extracellular acidosis is not clearly immunosuppressive, but can have both promoting and suppressive effects on different classes of immune cells"[57]. Our mechanism explains the apparent paradox of how the LN is able to host processes that underpin T-cells activation, while at the same time suppressing T-cells from invoking their effector functions while in residence. This physiological mechanism may, however, be exploited by tissues seeking to evade immune surveillance, such as solid tumours. In the case of tumours, however, acidity can be manipulated, as demonstrated by the efficacy of systemic buffers on improving T-cell checkpoint blockade therapy[7]. Tertiary lymphoid structures (TLS) are ectopic lymphoid-like organs found in nonlymphoid tissues, which develop under conditions of persistent chronic inflammation, such as in tumours, in autoimmune syndromes, and inflammatory disorders[58]. Some TLSs exist as sophisticated, segregated structures that bear resemblance to LNs[59]. It is plausible that these TLSs, sharing structural similarities with LNs, would also manifest an acidic pH, therefore locally inhibit T-cell-dependent immune functions. Accumulating evidence supports that TLSs are important in antitumoural immunity[60], therefore increasing T-cell function by selectively manipulating the pH of tumour-associated TLSs, may benefit immunotherapy.

## Methods

**Isolation and activation of T-cells.** Female B6 (C57BL/6), Pmel, OT-I, OT-II and TDAG8-knockout (TDAG8 KO) mice on the C57BL/6 background were bred and housed at the Animal Research Facility of the H. Lee Moffitt Cancer Center and Research Institute (Tampa, FL). Naïve T-cells were isolate from mice spleen using T-cell column (R&D system). T-cells from B6 mice and TDAG8-knockout mice were cultured in complete medium with 5 μg/mL plate-bounded anti-CD3 antibody and 2 μg/mL soluble anti-CD28 antibody for 48 h. Isolated Pmel, OT-I and OT-II T- cells are cultured in complete medium with 5 μg/mL gp-100 $_{25–33}$ peptide, 10 μg/mL OVA $_{SIINFEKL}$ peptide and 10 μg/mL OVA $_{323–339}$ peptide, respectively. IFN-gamma was measured by ELISA (BD Biosciences). All animal experiments were approved by the Institutional Animal Care and Use Committee and performed in accordance with the U.S. Public Health Service Policy and National Research Council Guidelines. Jurkat cells were maintained in RPMI-1640 medium with 5% FBS. Jurkat cells were stimulated with phorbol 12-myristate 13-acetate

(PMA, Cat#8139, Sigma–Aldrich) and phytohemagglutinin, M form (PHA-M, Cat# 10576015, Gibco) for 24 h.

**Animals**. All animals were maintained under Institutional Animal Care and Use Committee (IACUC) at H. Lee Moffitt Cancer Center. Eight-to ten-week old Balb/c, C57BL/6, and $nu/nu$ mice (male, 22–25 g) were purchased from The Jackson Laboratory and housed in ventilated isolette cages at ambient temperature and humidity with 12 h light dark cycles.

**LN lactate measurement**. Inguinal lymph nodes (LNs) excised from a consistent anatomical location were surgically remove from immunocompetent C57BL/6 (B6) or nude mice, weighted and flash frozen in liquid nitrogen immediately. Tissue was homogenized in 0.2 mL 80% methanol and the supernatants obtained after 10 min of centrifugation at $15,000 \times g$ were collected for biochemical analysis. Lactate concentration was measured by a fluorometric method using Lactate Assay Kit (BioVision, inc. Cat#K607).

**Seahorse measurements of metabolism**. Extracellular acidification rate (ECAR) and oxygen consumption rate (OCR) were measured by Seahorse XF96 Analyzer (Agilent). Cells were cultured with bicarbonate-free RPMI-1640 medium with 2 mM HEPES and 2 mM MES. The buffering capacity was determined to calculate the proton production rate (PPR).

**Flow cytometry**. Fresh isolated T-cells were activated at pHe 7.4 or pHe 6.6 for 72 h. Cells were collected and wash by PBS twice, then stained in FACS buffer with the following antibodies for flow cytometric analysis: CD3, CD4, CD8, CD44, and CD62L (see Supplementary Table S3 for antibody information). Live/Dead fixable near-IR (Invitrogen) was used to exclude dead cells before analysis. To analyze intracellular marker IFNγ, cells were incubated with 1 μL/mL GoldGiPlug (BD Bioscience) for 3 h, stained with surface marker and Live/Dead dye, fixed and permeabilized by Fixation/Permeabilization Solution Kit (BD Biosciences), and then stained with anti-IFNγ antibody. Samples are analyzed by LSR II Flow Cytometer (BD Biosciences). Multiple antibody lot numbers were used and each was validated by the flow cytometry core facility according to the manufacturer prior to used and titered for appropriate staining by us. In general, antibodies were used at a dilution of 1 ul per 100 ul staining buffer per $10^6$ cells.

**Antibodies**. Anti-pimonidazole antibody (#PAb2627, a rabbit polyclonal antibody) was purchased from Hyproxyprobe, Inc (Burlington, MA) and used at a 1:100 dilution; anti-CD3 antibody (#M3072, a rabbit monoclonal antibody) was purchased from Spring Bioscience Corp. (Pleasanton, CA) and used at a 1:100 dilution; anti-CD28 antibody (37.51, 16-0281-82) was purchased from Thermofisher (Waltham, MA) and used at a concentration of 1 ug/mL; anti-CD4 antibody (GK1.5, BE0003-1) was purchased from Bioexcell (Lebanon, NH) and used at a concentration of 3 ug/ul; anti-CD8 antibody (2.43, BE0061) was purchased from Bioexcell (Lebanon, NH) and used at a concentration of 3ug/ul.

**In vivo depletion of CD4 and CD8 T-cells**. C57B6 mice were injected IP with CD4 (GK1.5) and CD8 (2.43) depleting antibodies at a dosage of 300 ug/mouse for three consecutive days to initiate depletion. Depleted state was then maintained by additional dosing every 3 days until initiation of imaging studies. Depletion status was verified by flow cytometry on isolated lymph nodes and spleen of depleted and nondepleted mice.

**Cytokine beads array assay**. T-cells were activated for 48 h and restimulated at pHe 7.4 or pHe 6.6 for 24 h. Culture medium was collected for cytokine beads array analysis according to the manufacturer's manual (BioLegend). Briefly, 25 μL culture medium was sequentially mixed with antibody-conjugated beads, detection antibody and SA-PE. Washed samples were analyzed by flow cytometer.

**Cell proliferation assay**. Fresh prepared T-cells were washed by PBS twice and stained with 2 uM CellTrace Voilet (Invitrogen) in PBS for 10 min, and incubated in complete medium for another 20 min to quench residual dye. After two wash with complete medium, cells were activated at pHe 7.4 or pHe 6.6 for 72 h. After activation, cells were collected and stained with surface marker and live/dead dye for before analysis.

**Cytotoxicity assay**. For the cell lysis assay, the Xcelligence system (Roche Diagnostics) was used to monitor cellular events without incorporation of radioactive labels. Fifty microliter of complete media (CM) was added to 96XE-plates. Twenty thousand target cells (B16 or B16 pulsed with OVA peptide) were seeded into the wells of 96XE-Plates in 50 μL of CM and incubated on the Real Time Cell Analyzer overnight in a CO2 incubator to monitor cell adhesion and growth. Effector cells (OT-I T-cells) activated for 24 h with OVA peptide in media at pH 6.6 or pH 7.4 were added to plate at 25:1 ratio in a volume of 100 μL/well. Co-cultures were assessed by the system for 20 h. Results are expressed as percent lysis determined

from Cellular Index (CI) normalized as (nCI): % of lysis = [nCI (no effector) − nCI (effector)]/nCi (no effector) × 100.

**Phosphofructokinase-1 (PFK-1) activity assay**. T-cells from spleens of B6 mice were activated for 48 h and washed by PBS twice, and homogenized in M-PER Mammalian Protein Extraction Reagent (cat#78501, ThermoFisher) with Halt^TM Protease Inhibitor Cocktail (1:100, cat#87786, ThermoFisher). The supernatants, after 10 min of centrifugation at $15,000 \times g$, were collected for enzymatic analysis. Ten microlotre of supernatants were added to 2 mL of reaction buffer (50 mM HEPES, 1 mM ATP, 1 mM fructose-6-phosohate, 2 mM MgCl₂, 0.2 mM NADH, 1 U/mL aldolase, 5 U/mL triosephosphate isomerase, 1 U/mL α-glycerophosphate dehydrogenase, pH varied between 6.6 and 7.4) to initiate the reaction and the change in absorbance at 340 nm were measured spectrophotometrically every 10 s to calculated the enzyme activity.

**Immunochemistry**. Murine inguinal lymph nodes were surgically removed from C57BL/6 mice, fixed in formalin and paraffin embedded. Slides were prepared with 4-μm thick tissue slices and stained using a Ventana Discovery XT automated system (Ventana Medical Systems) as per manufacturer's protocol with proprietary reagents. The primary antibodies were used to detect pimonidazole (1:100, Hyproxyprobe #PAb2627) and CD3 (1:100, Spring Bioscience #M3072) expression. Slides were incubated with Ventana OmniMap Secondary Antibody followed by Ventana ChromoMap kit to detect the proteins staining and then slides were counterstained with Hematoxylin.

**Superfusion**. Superfusion experiments were performed in a plastic chamber (Pecon, TempController 2000-2) supplied by a solution line with a switcher that changed between one of two lines (the other being diverted to the waste bottle). The plastic chamber was mounted on a confocal microscope and heated to 37 °C by small scale temperature incubator (The Cube Life Imaging Services). Solution exchange was attained with a time constant of 2.6 s. Solution flows were 2–4 ml/min.

**Solutions and media**. (i) Solutions for seahorse experiments: 2 mM HEPES, 2 mM MES, 5.3 mM KCl, 5.6 mM Na-Phosphate, 11 mM glucose, 133 mM NaCl, 0.4 mM MgCl₂, 0.42 mM CaCl₂, titrated to given pH with NaOH. For reduced Cl⁻ experiments, 133 mM NaCl was replaced with 133 Na-Gluconate and MgCl₂ and CaCl₂ were raised to 0.74 and 1.46 mM, respectively, to account for gluconate-divalent binding. Amount of dilute HCl or NaOH added to medium to reduce pH to target level was determined empirically. Solutions for pH measurements under superfusion: For pH 7.4, 133 mM NaCl, 5.3 mM KCl, 10 mM Glucose, 1 mM CaCl₂, 1 mM MgCl₂, 22 mM NaHCO₃. For lower pH, NaHCO₃ was reduced (compensated by NaCl) to attain a target pH, according to the Henderson Hasselbalch equation (pH = 6.15 + log([HCO₃⁻]/[CO₂]), where [CO₂] is 1.2 mM for 5%. All solutions were bubbled in 5% CO₂. (iii) Solutions for Ca²⁺ imaging: For pH 7.4, 133 mM NaCl, 5.3 mM KCl, 0.8 mM MgCl₂, 0.9 mM Na-Phosphate, 22 mM NaHCO₃ and either 1.8 mM CaCl₂, 0.5 mM CaCl₂ or 0.5 mM EGTA. For pH 6.6, NaHCO₃ was reduced to 2.75 mM and NaCl raised accordingly. All solutions were bubbled with 5% CO₂/balanced air. (iv) Calibration solutions for nigericin: 145 mM KCl, 1 mM MgCl₂, 0.5 mM EGTA, 10 mM HEPES, 10 mM MES and pH adjusted with NaOH to required level.

**Confocal imaging**. Imaging was performed on an SP5 system (Leica Microsystems). Cellular measurements were performed with an oil-immersion ×63 objective and intravital microscopy was performed with a dry ×1.6 or ×10 objective. The following excitation (ex) and emission (em) wavelengths were used: dextran-conjugated cSNARF1 (cat#D3304, Invitrogen): 514 nm ex, 580/640 nm em (during lymph node imaging); Hoechst 34580 (cat#H21486, ThermoFisher): 405 nm ex, 420 nm; FuraRed: 488 nm ex, 585/685 nm; pHLIP-Cy5.5: 633 nm ex, >700 nm. Settings were optimized to obtain maximal quality under the constraints of temporal resolution.

**Lymph node imaging**. The inguinal lymph node is exposed for imaging on the confocal microscope by performing a midline incision to separate the skin from the peritoneum. The skin is pinned down and the excess fat around the inguinal lymph node is carefully removed with sterile Dumont #5 forceps. Once cleared and exposed, a 3D printed window chamber, with a 12 mm in diameter window, is placed over the lymph node area. The chamber is secured using a tissue adhesive (3 M Vetbond #1469SB) and 12 mm Micro coverslip (cat# 72226-01 Electron microscopy sciences). We ensure that the lymph node region underneath the coverslip does not dryout by injecting 200 μl of 1× PBS. Mice were kept under anesthesia (1.5% isofluorane) throughout the surgical procedure and in a 37 °C warming chamber during imaging. To measure pH in the inguinal lymph node, mice were injected with dextran-conjugated cSNARF1 (cat# D3304, Invitrogen) at a concentration of 20 mg/ml in 100 μl via their tail-vein or footpad. To determine whether inflammation or buffering would change the pH of the lymph node microenvironment, mice were treated with lipopolysaccharide (LPS, cat# L3012, Sigma–Aldrich) at a concentration of 1000 ng/kg (i.p. injection) for 48 h or

provided mice with 400 mM $NaHCO_3$ ad libitum for 9–10 days, respectively, before imaging the mice with dextran-conjugated cSNARF1.

**Image analysis.** Cytoplasmic pH was measured by gating pixels according to a threshold level of Hoechst signal within cSNARF1-positive pixels. Fluorescence at 580 and 640 nm was averaged, background offset and ratioed for each particle representing a cell. For time course experiments, pHi was probed in the entire cell to allow for faster acquisition rates. Buffering capacity was measured from the change in weak acid/base concentration, assuming passive equilibration, and the pH change. Transmembrane acid-base fluxes were therefore calculated as the product of pH change and buffering capacity. Cell surface area/volume ratio assumed spherical symmetry i.e. 3/radius. For intravital microscopy, image montages were constructed with in-house software that aligned fluorescence or anatomical landmarks.

**CEST imaging.** Magnetic resonance data were acquired with a 7T horizontal Bruker scanner equipped with nested 205/120/HDS gradient insert and a bore size of 310 mm. A 35 mm Litzcage coil (Doty Scientific) was used to carry out all experiments. Before imaging, animals were placed in an induction chamber and anesthetized with 3% isoflurane delivered in 1.5 litre/min oxygen ventilation. After complete induction, animals were restrained in a custom-designed holder and inserted into the magnet while constantly receiving isoflurane (1–3%) within the 0.6 litre/min oxygen ventilation. Body temperature (37° ± 1 °C) and respiratory functions were monitored continuously (SAII 177 System) during the experimental time. Coronal $T_2$-weighted fast spin-echo multislice images were acquired with TE/TR [echo time/repetition time] = 31 ms/2271 ms, field of view (FOV) = 80 × 30 mm², matrix = 256 × 96, yielding a spatial in-plane resolution of 312 μm and with slice thickness of 1.5 mm. These images were used as anatomical reference for the pH map. MRI-CEST pH images were acquired by adapting a previously described protocol[29]; TE/TR = 10 ms/10 s, the saturation used was 3 μT during 5 s, same FOV than T2w images but matrix = 171 × 64. Only one slide was imaged containing the inguinal lymph node. Animals were injected with ISOVUE370 (Bracco Imaging, Milano, Italy) at 300 ul iv bolus injection; followed by an i.v. infusion of 300 μl/h. To create the CEST maps, an in-lab designed Matlab code was used. The pH values were calculated after a calibration curve done in the same system with 20 mM ISO-VUE370 phantoms titrated at several pH values in the range 5.5–8.

**Statistics.** All statistical tests had a significance level of 5% in a two-tailed test. For comparisons between two samples, a t-test was used. For more than two samples, a one-way ANOVA with multiple comparisons was used.

**Reporting summary.** Further information on research design is available in the Nature Research Reporting Summary linked to this article.

## Data availability
High-resolution image data are available upon request from either of the corresponding authors (PS or RJG). All other relevant data are available in the article, supplementary information, or from the corresponding authors (PS or RJG) upon reasonable request. Source data are provided with this paper.

## Code availability
Custom code was developed for image processing of data in Fig. 1g–i, and for steady-state modeling of Fig. 2g, h. It allowed ad hoc image stitching, curve fitting, etc. It is available upon request to corresponding author PS.

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

## Acknowledgements

This work was supported by U.S. Department of Health & Human Services | NIH| National Cancer Institute (NCI) - R01 CA077575 [Gillies]. U.S. Department of Health & Human Services | NIH| National Cancer Institute (NCI) - U54 CA193489 [Gillies]. U.S. Department of Health & Human Services | NIH| National Cancer Institute (NCI) - P30 CA076292 [Gillies]. Fulbright Association - 0001 [Swietach]. European Research Council "SURVIVE" #723997 [Swietach]. U.S. Department of Health & Human Services |NIH| National Cancer Institute (NCI) - F99 CA234942 [Russell]. U.S. Department of Health & Human Services | NIH| National Cancer Institute (NCI) - R01 GM073857 [Reshtenyak/Andreev]. Associazione Italiana Ricerca Cancro |AIRC| MFAG 2017 - ID. 20153 [Longo]. China Scholarship Council 201706325051 [Wu]. Natural Science Foundation of Zhejiang Province, China LY17H160036 [Wu]. U.S. Department of Health & Human Services |NIH| National Cancer Institute (NCI) - R01 CA239219 [Gillies/Pilon Thomas].

## Author contributions

Conception or design of the work: H.W., V.E., D.A., A.I-H., K.L., S.P-T., P.S., R.J.G. Data collection: H.W., V.E., M.B., A. E-K., S.R., D.A., A.I-H., T.N., A.M., O.A., K.L., M.D., K.K., S.R.P., P.E-N., P.S. Data analysis and interpretation: H.W., V.E., M.B., D.L.L., Y.R., O.A., P.E-N, S.P-T., P.S., R.J.G. Drafting the article: S.P-T., P.S., R.J.G. Critical revision of the article. H.W., V.E., M.B., D.A., A.I-H., Y.R., O.A., K.L., S.P-T., P.S., R.J.G. Final approval of the version to be published. H.W., V.E., M.B., Y.R., S.P-T., P.S., R.J.G.

## Competing interests

R.J.G. has research support form Helix Biopharma, who makes acid pH targeting agents. These agents were not used in the current work. O.A.A. and Y.K.R. are founders of pHLIP, Inc. They have shares in the company, but the company did not fund any part of the work reported in this paper, which was carried out in their academic laboratories. No other authors report competing interests.
