## [Peer Review File · Nature Communications]

Reviewers' comments:

Reviewer #1 (Remarks to the Author): expertise in imaging metabolism

This paper describes a comprehensive and sophisticated study which shows that the low pH in lymph nodes (LN), while not inhibiting T cell activation does inhibit the release of some cytokines, protecting the LN from premature release of these cytokines. The low pH is shown to be mediated by glycolytic lactic acid production by the T cells, which is feedback inhibited by lactate and H⁺ ions in the extracellular space. The mechanism by which inhibition of glycolytic flux inhibited cytokine release was not determined, although some potential mechanisms were ruled out. This is a publishable story. I have only one, relatively minor, criticism.

The authors argue that the failure of LPS treatment to alter LN pH is consistent with a robust pH stat mechanism, whereby an enlargement of the LN does not disrupt this pH stat. However, no evidence is provided that the LNs are enlarged. Also, presumably, LPS would lead to T cell activation (and activation of other immune cells in the LN?), increasing glycolytic flux and lowering pH, offsetting any increase in LN volume? In summary, I did not find this compelling in vivo evidence for their pH stat mechanism. Similarly, they argue that the failure of their bicarbonate infusion experiment to raise LN pH was also evidence for the pH stat mechanism in vivo. To provide conclusive proof for this they would need to determine the effect of bicarbonate on glycolytic flux in the lymph node, according to their mechanism bicarbonate should stimulate glycolytic flux. I accept that these are difficult experiments to do (maybe micro dialysis in animals administered with isotope-labelled glucose?) but without them one can propose alternative explanations for their results without invoking the pH stat mechanism, for example that insufficient bicarbonate gets to the LN and therefore cannot elevate pHe. What happens to pHe in the LN of nude mice, which have no T cells and reduced acidity, following bicarbonate administration? This would be a simpler experiment to do than trying to measure glycolytic flux in the LN

Reviewer #2 (Remarks to the Author): expertise in T cell signalling and metabolism

The authors described an acidic niche within the immune system that regulates T cell activation. While the model is interesting, the current data presented are not sufficient to support the models, and more substantial results are required.

1. The authors proposed a key role of T cells in affecting the acidic environment in the lymph node. Although they used athymic mice to evaluate the involvement of T cells, this system is not conclusive as it can be influenced by long-term effects of T cell deficiency. The authors should apply acute depletion of T cells to test the direct role of T cells. Also, they need to examine organs other than the lymph node to ascertain the specificity of the observed phenomena.
2. The authors proposed the importance of lactate produced by activated T cells in affecting the acidic niche, but the data are entirely based on in vitro observations. It is important to note that antigen-specific T cells represent a very small subpopulation of T cells, and it is unclear if lactate produced by such antigen-specific T cells in an immune response in vivo is sufficient to alter the pH of the lymph node. The authors need to rigorously examine the role of activated T cells in vivo.
3. The authors proposed that the effect of dampened T cell activation from the acidic niche is to prevent tissue damage of the host organ, but there is no data presented. These results are essential to establish the physiological relevance of the model.

Reviewer #3 (Remarks to the Author):expertise in lymphatic system and mathematical modelling

Regulation of immunity is still poorly understood, and its importance has been underscored by recent advances in tumor immunotherapies. The authors present a study analyzing lymph node pH and its effect on T cell biology and metabolism. They conclude that T cells create and maintain an acidic niche in the lymph node that is self-regulating and which suppresses effector cell function.

The presented analyses of T cell glycolysis, intracellular and extracellular pH, and cytokine production –both in vitro and in vivo – are thorough and convincing. It is clear that pH can affect the phenotype of these cells. However, the central thesis of the work – that effector cell function is limited in lymph nodes, but enabled upon exodus – is not consistent with known regulation of immunity, and conflicts with other data. Most importantly, the authors seem to confound effector cell activation and proliferation (which should happen in the lymph node) with effector cell cytotoxic activity (which should happen in the target tissue). This is a fundamental flaw in rationale that draws into question the relevance of the work.

Specific comments:

1. Many tissues have baseline acidic pH, and wound beds are often acidic. Evolutionarily, it would be disadvantageous to disable the adaptive immune response in any acidic environment. There is actually evidence that acidic wounds heal faster. Can the authors reconcile this with their findings?

2. The authors show that acidity decreases T cell proliferation and cytokine production. But these are the main activities that are supposed to happen in the lymph node. Antigen presentation leads to activation and effector cell proliferation locally, before the cells leave the node. There are also critical paracrine effects of the cytokines that would be prevented by the authors' proposed mechanism. For example, helper T cells activated in the node produce cytokines that activate CD8 T cells and B cells in the same node. In this way, the lymph node is a factory of effector cells and antibodies that, if inhibited by acidic pH, would be expected to thwart systemic immunity. These issues are not discussed in the manuscript.

3. Similarly, the authors suggest that low pH in the node promotes differentiation to a memory phenotype in T cells. However, memory T cells are thought to form in the target tissue at the end of the immune response. These cells should not be traveling back to the lymph node before differentiating into memory cells.

The most important aspect of effector cell function is not analyzed here. While cytokine production is important for effector cell activation and proliferation, cell killing determines the efficacy of the immune response. It would make sense that cytotoxic functions are suppressed in the nodes, but this is not discussed or analyzed.

4. The authors write "T-cells activated by antigen-presenting cells (e.g. DCs) produce an acidic environment...." This should be shown in vivo, in addition to the in vitro experiments. It is not clear that the lymph nodes in Figure 1 were involved in antigen -induced activation. If not, then the results show that even quiescent nodes are acidic.

5. The authors write that "once outside the acidic LN, effector functions of egressing T-cell become rapidly uninhibited." Do the authors think this happens in the blood stream? Or during transit through the lung? Or only at the infection site? It seems like we would need other control mechanisms to make sure it happens only in the target tissue.

6. The success of anti-PD1 and anti_PDL1 immunotherapies in cancer shows that it is possible to activate effector cell function in tumors, presumably even in the acidic tumor environment. Can the authors reconcile this discrepancy?

Minor comments:

7. The author should provide more information about the mechanism of how pHLIP responds to pH.
8. Why was the pimonidazole injected IP rather than into the footpad or i.v.?

Reviewers' comments:

Reviewer #1 (Remarks to the Author): expertise in imaging metabolism

This paper describes a comprehensive and sophisticated study which shows that the low pH in lymph nodes (LN), while not inhibiting T cell activation does inhibit the release of some cytokines, protecting the LN from premature release of these cytokines. The low pH is shown to be mediated by glycolytic lactic acid production by the T cells, which is feedback inhibited by lactate and H⁺ ions in the extracellular space. The mechanism by which inhibition of glycolytic flux inhibited cytokine release was not determined, although some potential mechanisms were ruled out. This is a publishable story. I have only one, relatively minor, criticism.

- The authors argue that the failure of LPS treatment to alter LN pH is consistent with a robust pH stat mechanism, whereby an enlargement of the LN does not disrupt this pH stat. However, no evidence is provided that the LNs are enlarged. Also, presumably, LPS would lead to T cell activation (and activation of other immune cells in the LN?), increasing glycolytic flux and lowering pH, offsetting any increase in LN volume? In summary, I did not find this compelling in vivo evidence for their pH stat mechanism. We have repeated the LPS experiments and report a 50% increase in area measured in the x-y plane (P=0.002; now added as Fig S5). The LN is undoubtedly a complex system, but the observation that its pHe remains constant after LPS-stimulated T-cell proliferation and organ growth suggests that the mechanism underpinning pHe-control is regulated towards a target pH. This would be characteristic of a feedback loop, whereby acid-production (glycolysis) is controlled by the degree of acid-accumulation (i.e. pH), so that once pHe=pH_{stat}, further acid production is shut down. It is, therefore, expected and consistent with our model that pHe is unchanged in response to LPS. The only way to break this loop would be to eliminate the source of acid (T-cells) or manipulate the pH sensor, which may be experimentally intractable.*

- Similarly, they argue that the failure of their bicarbonate infusion experiment to raise LN pH was also evidence for the pH stat mechanism in vivo. To provide conclusive proof for this they would need to determine the effect of bicarbonate on glycolytic flux in the lymph node, according to their mechanism bicarbonate should stimulate glycolytic flux. I accept that these are difficult experiments to do (maybe micro dialysis in animals administered with isotope-labelled glucose?) but without them one can propose alternative explanations for their results without invoking the pH stat mechanism, for example that insufficient bicarbonate gets to the LN and therefore cannot elevate pHe. What happens to pHe in the LN of nude mice, which have no T cells and reduced acidity, following bicarbonate administration? This would be a simpler experiment to do than trying to measure*

glycolytic flux in the LN. We agree that further experimental evidence would strengthen our hypothesis. According to our hypothesis, an increase in buffering capacity is expected to stimulate acid production, and hence lead to more lactate retention in the LN. We tested this in mice treated with oral *ad lib* bicarbonate for 2 weeks prior to harvesting the LNs and measuring [lactate].

Control conditions:
System reaches pH_{stat} in equilibrium [lactate]

Enhanced buffering:
Higher buffering allows greater release of acid.
System reaches comparable pH_{stat} at higher [lactate]

In these animals, the pH of LNs were unaffected (i.e. remaining near the pH_{stat}), yet [lactate] in both axillary and inguinal nodes increased by ~50% showing that increased buffering allowed for a greater acid-production. These data are now discussed in text and are provided as (Fig S6A-B).

Reviewer #2 (Remarks to the Author): expertise in T cell signaling and metabolism

The authors described an acidic niche within the immune system that regulates T cell activation. While the model is interesting, the current data presented are not sufficient to support the models, and more substantial results are required.

1. The authors proposed a key role of T cells in affecting the acidic environment in the lymph node. Although they used athymic mice to evaluate the involvement of T cells, this system is not conclusive as it can be influenced by long-term effects of T cell deficiency. The authors should apply acute depletion of T cells to test the direct role of T cells. Also, they need to examine organs other than the lymph node to ascertain the specificity of the observed phenomena. To address this, C57/Bl6 mice were lymphodepleted with anti-CD4 and anti-CD8 antibodies (300 μg/mouse)

administered I.P. once per day for 3 days and then on every 3rd day to maintain depletion. LNs and spleens were harvested to quantify degree of depletion. These data show a depletion of CD3+ T cells within 10 days of initiation by ~80% in the spleen and inguinal LN (**Fig S3**).

Following these control experiments, LNs of individual control and depleted mice were imaged 24 hr after injection of pHLIP. Representative pHLIP images are provided below and in **Fig S2D**. A quantification of these data has been added to **Figure 1F**. We observed that an acute depletion produced pHLIP signal that was between levels in control and nude mice. This was expected, because acute depletion with anti-CD4 + anti-CD8 antibodies left a residual population of T cells, which are able to contribute some degree of acidification. After imaging, LNs were removed and % CD3 cells was quantified (provided in supplemental **Table S1**) and showed that depletion ranged from 74-98%. Thus, the pHLIP signal was related to the number of T-cells, in a dose-dependent manner.

2. *The authors proposed the importance of lactate produced by activated T cells in affecting the acidic niche, but the data are entirely based on in vitro observations. It is important to note that antigen-specific T cells represent a very small subpopulation of T cells, and it is unclear if lactate produced by such antigen-specific T cells in an immune response in vivo is sufficient to alter the pH of the lymph node. The authors need to rigorously examine the role of activated T cells in*

in vivo. As first step in addressing this difficult question, we have quantified the absolute number of T cells in LNs of control mice (**Fig S8**). The count of 2 million per LN is consistent with literature.

The *in vitro* experiment attempted, in as much as is feasible, to mimic the chemical milieu of the LN *in vivo*. High-resolution MRI have shown that the volume of inguinal and axillary LNs of C57Bl6 mice were 2.5±0.3 μ l (1), giving a CD3+ T-cell density of 800 million/mL. The highest density of cells we can achieve reliably for *in vitro* measurements in the IBIDI chambers was 15 million/mL. Thus, it was necessary to reduce buffering capacity, by switching to low-buffered 2mM HEPES+2mM MES; buffering capacity of 3.78 mM/pH over the pH range 6.0-7.5. From the data in **Fig 2E**, we could calculate that the rate of acid production by activated and naïve T cells was 0.3 and 0.03 nmoles/min per 10⁶ cells, respectively. Even if all T-cells were naïve, the mass of CD3+ cells would generate 1.45 mmol/L of H⁺ per hour, which would double if 12% of the T-cell population became activated. Assuming that capillary wash-out is limited, these rates of acid-loading could reduce pH within <16 hours to levels as low as 6.2.

Indeed, even at the relatively low densities used *in vitro*, the T cells acidified to a set point of pH 6.2 within 2 hours (activated T cells) or 10 hours (naïve T cells). Notably, the *in vivo* dextran-SNARF pH measurements also showed a pHe of circa 6.2 in the most acidic sub-regions. While we were initially surprised that pHe was so low, the data do indicate that the niche of LNs can be quite acidic. This is consistent with a feedback circuit, whereby acid is produced continually, until its production is inhibited by a critically low pH (here, ~6.2). Provided the washout by capillary perfusion is limited, this equilibrium-point will be reached even if the fraction of activated T-cells is low; the only difference being how *long* it takes to attain steady-state.

3. *The authors proposed that the effect of dampened T cell activation from the acidic niche is to prevent tissue damage of the host organ, but there is no data presented. These results are essential to establish the physiological relevance of the model. This is extremely difficult to accomplish with certainty in vivo, as any mechanism to reduce acidity would also affect metabolism, which could*

have its own detrimental effects. To respond to this concern as well as we could, we performed a cytotoxicity assay using Xcelligence (Roche) microplate reader seeded with B16 cells expressing ovalbumin, and treated with CD8+ T-cells from OTII mice that have an engineered anti ovalbumin TCR. Within 48 hr, 46% of the B16 OVA cells at pH 7.4 had been lysed compared to less than 5% at pH 6.8, indicating that their cytolytic activity of T cells is dramatically inhibited by acidity.

To prove this *in vivo* would be difficult. However, we have been able to develop a case from literature, which is described below.

Our argument is based on the observation that immune cells are differentially affected by a reduction in pHe, as we have elaborated in a recent review (2). Under acidic conditions, dendritic cells express higher levels of maturation markers, have improved antigen presentation, and are better activators of T cells (3,4). In contrast, T cells have *reduced* activation states and suppressed cytokine secretion (5). While it may simply be that acidic conditions are detrimental to T cell processes and helpful to DC processes, we propose that this dichotomy reflects an evolutionary outcome and that T cell inhibition at low pH is not a flaw in the system, but instead serves an immunoregulatory function. Therefore, there would be instances where low pH is exploited by the host tissue to suppress an inappropriate T cell mediated immune response. During an immune response, the T cell compartment of the LN houses a dense population of T cells, which expands following TCR ligation (6). **It is known that aberrant activation of densely packed T cells can induce immunopathological responses in both lymphoid and non-lymphoid tissues, and many checkpoints are in place to prevent overactive lymphocytes (7-9).** It is intuitive to suspect that lymph nodes rely on protective measures to avoid off-target and inflammation-driven pathology.

Lymph node structure plays a critical role in its function and loss of this architecture can diminish lymph node output (10,11). Overactive immune responses can impact lymph node structure and function. Persistent immune activation within lymphoid tissue, as seen with human immunodeficiency virus (HIV), results in collagen deposition and lymph node fibrosis, often restricted to the T cell zone, leading to diminished lymph node function and reduced peripheral T cell numbers (11-13). Furthermore, a link between increased immune activation and diminished lymph node function also exists in an HIV-negative population. Kityo et al compared HIV-negative Ugandans with groups of HIV-negative participants from Minnesota and Georgia. For unclear reasons, the Ugandan cohort had increased systemic immune activation. There was an increase in immune activation within lymph node tissues of the Ugandan cohort, and this corresponded to increased lymph node fibrosis, diminished T cell numbers and impaired vaccine responses (12).

Additionally, high levels of cytokines accumulating within the T cell zone of the lymph node could potentially have detrimental effects on immune cells as well as cells residing in the lymph node. For example, IFN-gamma, a pleiotropic cytokine that is post-translationally inhibited by low pH, has diverse effects that are often dose- and context-dependent (14). IFN-gamma regulates T cell polarization, induces apoptosis, inhibits hematologic and lymphatic vessel formation and regulates T cell homeostasis (15). Preventing an unregulated accumulation of IFN-gamma within the T cell compartment is likely beneficial to the immune response.

For space reasons we have distilled this argument to the bolded points, and the 2nd paragraph of the introduction now reads: *"Given the exquisite pH-sensitivity of cytokine release^{3,4,6,7}, LN acidity may have physiological consequences. It would be advantageous to refrain from secreting inflammatory cytokines to avoid inflicting damage to the host organ. Aberrant activation of densely packed T cells can, for example, induce immunopathological responses in both lymphoid and non-lymphoid tissues and, for that reason, many checkpoints are in place to prevent overactive lymphocytes in these organs⁸⁻¹⁰. While these checkpoints are active under physiological conditions, a pathologically overactive immune response can negatively impact lymph node structure and function. Persistent immune activation within lymphoid tissue, as seen with human immunodeficiency virus (HIV), results in collagen deposition and lymph node fibrosis, often restricted to the T cell zone, leading to diminished lymph node function and reduced peripheral T cell numbers¹¹⁻¹³. Additionally, high levels of cytokines accumulating within the T cell zone would have detrimental effects on the acquisition of adaptive immunity. For example, IFN γ , whose expression is potently inhibited at low pH, alters T-cell polarization and homeostasis, can induce apoptosis, and inhibit lymphangiogenesis¹⁴⁻¹⁶. However, without direct measurements of pH in intact LNs, the physiological significance of this postulated regulatory influence is untested."*

Reviewer #3 (Remarks to the Author): expertise in lymphatic system and mathematical modelling

Regulation of immunity is still poorly understood, and its importance has been underscored by recent advances in tumor immunotherapies. The authors present a study analyzing lymph node pH and its effect

on T cell biology and metabolism. They conclude that T cells create and maintain an acidic niche in the lymph node that is self-regulating and which suppresses effector cell function.

The presented analyses of T cell glycolysis, intracellular and extracellular pH, and cytokine production both in vitro and in vivo are thorough and convincing. It is clear that pH can affect the phenotype of these cells. However, the central thesis of the work - that effector cell function is limited in lymph nodes, but enabled upon exodus - is not consistent with known regulation of immunity, and conflicts with other data. Most importantly, the authors seem to confound effector cell activation and proliferation (which should happen in the lymph node) with effector cell cytotoxic activity (which should happen in the target tissue). This is a fundamental flaw in rationale that draws into question the relevance of the work.

Specific comments:

1. Many tissues have baseline acidic pH, and wound beds are often acidic. Evolutionarily, it would be disadvantageous to disable the adaptive immune response in any acidic environment. There is actually evidence that acidic wounds heal faster. Can the authors reconcile this with their findings? We agree that wound beds are acidic. Indeed, an acid pH promotes wound healing by its anti-microbial activity and an alkaline pH inhibits wound healing (16). Acidity does not negatively affect innate immunity which is critically important in the acute phase of wound healing. We, and others, have shown that acidity promotes macrophages to assume a tissue-remodeling (M2) phenotype (17). However, wound healing does not involve adaptive immunity, nor does it require a T-cell response. Thus, we consider these observations to be internally consistent, and not contradicting the canon.
2. The authors show that acidity decreases T cell proliferation and cytokine production. But these are the main activities that are supposed to happen in the lymph node. Antigen presentation leads to activation and effector cell proliferation locally, before the cells leave the node. There are also critical paracrine effects of the cytokines that would be prevented by the authors' proposed mechanism. For example, helper T cells activated in the node produce cytokines that activate CD8 T cells and B cells in the same node. In this way, the lymph node is a factory of effector cells and antibodies that, if inhibited by acidic pH, would be expected to thwart systemic immunity. These issues are not discussed in the manuscript. Our data are not suggesting that acidity decreases T cell proliferation. Indeed, Fig S11D showed that proliferation of activated CD4+ and CD8+ T cells was only modestly affected by acidity. To investigate this with more sensitive tools, we performed [³H]thymidine incorporation assays of naïve (CM), and peptide- or CD3-activated pMel, OT-I and OT-II cells. As shown in the figure below (provided for review), while there was more incorporation at pH 7.4, it was not substantial, consistent with the flow data of S11D.

Proliferation

3. Similarly, the authors suggest that low pH in the node promotes differentiation to a memory phenotype in T cells. However, memory T cells are thought to form in the target tissue at the end of the immune response. These cells should not be traveling back to the lymph node before differentiating into memory cells. The most important aspect of effector cell function is not analyzed here. While cytokine production is important for effector cell activation and proliferation, cell killing determines the efficacy of the immune response. It would make sense that cytotoxic functions are suppressed in the nodes, but this is not discussed or analyzed. We agree that our observation regarding memory T cell conversion would require more data to document that it is occurring in LNs. Hence, we have removed this from the paper, as it is not central to the thesis and is somewhat a distraction. Regarding cytotoxicity, we performed Xcelligence cytotoxicity assays and showed that acid pH significantly inhibited cell killing (please see Reviewer 2 comment 3)
4. The authors write “T-cells activated by antigen-presenting cells (e.g. DCs) produce an acidic environment” This should be shown *in vivo*, in addition to the *in vitro* experiments. It is not clear that the lymph nodes in Figure 1 were involved in antigen-induced activation. If not, then the results show that even quiescent nodes are acidic. Our data do show that LNs, at rest, harbor niches that are significantly acidic (intravital imaging of pH-LIP in anesthetized mice). Even in the absence of a fulminant infection, T cells in LNs have a background resting activation state. Our *in vitro* data and mathematical models, informed by *ex vivo* analyses of resident T-cells, would indicate that there is sufficient metabolic activity to reach this level of acidity. (Please also see Reviewer #2, response 2).
5. The authors write that “once outside the acidic LN, effector functions of egressing T-cell become rapidly uninhibited.” Do the authors think this happens in the blood stream? Or during transit through the lung? Or only at the infection site? It seems like we would need other control mechanisms to make sure it happens only in the target tissue. What we have shown *in vitro* (Figure 4C) is that activated cells rapidly elaborate cytokines once they are returned to an environment

of mildly alkaline pH. It is reasonable to assume, therefore, that (in the absence of other inhibitory triggers) T cells will be active shortly after emerging from the acidic LN regions.

6. *The success of anti-PD1 and anti-PDL1 immunotherapies in cancer shows that it is possible to activate effector cell function in tumors, presumably even in the acidic tumor environment. Can the authors reconcile this discrepancy?* We do not believe that there is a discrepancy. Indeed, only ~20% of melanoma and lung cancer patients experience a durable clinical benefit of immune checkpoint blockade. There are multiple immune modulators, including *inter alia* MDSCs, IDO/kynurenine, etc., and we and others have shown that acidic pH is also a potent immune modulator. We have refrained from discussion of tumors or anti-cancer therapies in this manuscript as we feel it would be distracting from the major take-away lesson.

Minor comments:

7. *The author should provide more information about the mechanism of how pHLIP responds to pH.* We have added additional references describing the mechanism of action (18-20).
8. *Why was the pimonidazole injected i.p. rather than into the footpad or i.v.?* We have been using the same tested protocol for pimonidazole staining for over a decade. This protocol was optimized for tumor labeling and we determined that i.p. was as good as i.v. for tissue distribution but that it could be administered more reproducibly. We have never performed footpad of pimo as we saw no reason to pursue this, given the quality of i.p. delivery.

ADDITIONAL SUPPORTING REFERENCES

1. Economopoulos V, Noad JC, Krishnamoorthy S, Rutt BK, Foster PJ. Comparing the MRI appearance of the lymph nodes and spleen in wild-type and immuno-deficient mouse strains. *PLoS One* **2011**;6(11):e27508 doi 10.1371/journal.pone.0027508.
2. Damgaci S, Ibrahim-Hashim A, Enriquez-Navas PM, Pilon-Thomas S, Guvenis A, Gillies RJ. Hypoxia and acidosis: immune suppressors and therapeutic targets. *Immunology* **2018**;154(3):354-62 doi 10.1111/imm.12917.
3. Vermeulen M, Giordano M, Trevani AS, Sedlik C, Gamberale R, Fernandez-Calotti P, *et al.* Acidosis improves uptake of antigens and MHC class I-restricted presentation by dendritic cells. *J Immunol* **2004**;172(5):3196-204 doi 10.4049/jimmunol.172.5.3196.
4. Martinez D, Vermeulen M, von Euw E, Sabatte J, Maggini J, Ceballos A, *et al.* Extracellular acidosis triggers the maturation of human dendritic cells and the production of IL-12. *J Immunol* **2007**;179(3):1950-9 doi 10.4049/jimmunol.179.3.1950.
5. Erra Diaz F, Dantas E, Geffner J. Unravelling the Interplay between Extracellular Acidosis and Immune Cells. *Mediators Inflamm* **2018**;2018:1218297 doi 10.1155/2018/1218297.
6. Mempel TR, Henrickson SE, Von Andrian UH. T-cell priming by dendritic cells in lymph nodes occurs in three distinct phases. *Nature* **2004**;427(6970):154-9 doi 10.1038/nature02238.
7. Pilli D, Zou A, Tea F, Dale RC, Brilot F. Expanding Role of T Cells in Human Autoimmune Diseases of the Central Nervous System. *Front Immunol* **2017**;8:652 doi 10.3389/fimmu.2017.00652.
8. Skapenko A, Leipe J, Lipsky PE, Schulze-Koops H. The role of the T cell in autoimmune inflammation. *Arthritis Res Ther* **2005**;7 Suppl 2:S4-14 doi 10.1186/ar1703.
9. Walter U, Santamaria P. CD8+ T cells in autoimmunity. *Curr Opin Immunol* **2005**;17(6):624-31 doi 10.1016/j.coi.2005.09.014.
10. Schacker TW, Brenchley JM, Beilman GJ, Reilly C, Pambuccian SE, Taylor J, *et al.* Lymphatic tissue fibrosis is associated with reduced numbers of naive CD4+ T cells in human immunodeficiency virus type 1 infection. *Clin Vaccine Immunol* **2006**;13(5):556-60 doi 10.1128/CVI.13.5.556-560.2006.
11. Schacker TW, Nguyen PL, Beilman GJ, Wolinsky S, Larson M, Reilly C, *et al.* Collagen deposition in HIV-1 infected lymphatic tissues and T cell homeostasis. *J Clin Invest* **2002**;110(8):1133-9 doi 10.1172/JCI16413.

12. Kityo C, Makamdop KN, Rothenberger M, Chipman JG, Hoskuldsson T, Beilman GJ, *et al.* Lymphoid tissue fibrosis is associated with impaired vaccine responses. *J Clin Invest* **2018**;128(7):2763-73 doi 10.1172/JCI97377.
13. Zeng M, Smith AJ, Wietgreffe SW, Southern PJ, Schacker TW, Reilly CS, *et al.* Cumulative mechanisms of lymphoid tissue fibrosis and T cell depletion in HIV-1 and SIV infections. *J Clin Invest* **2011**;121(3):998-1008 doi 10.1172/JCI45157.
14. Sa Q, Woodward J, Suzuki Y. IL-2 produced by CD8+ immune T cells can augment their IFN-gamma production independently from their proliferation in the secondary response to an intracellular pathogen. *J Immunol* **2013**;190(5):2199-207 doi 10.4049/jimmunol.1202256.
15. Tewari K, Nakayama Y, Suresh M. Role of direct effects of IFN-gamma on T cells in the regulation of CD8 T cell homeostasis. *J Immunol* **2007**;179(4):2115-25 doi 10.4049/jimmunol.179.4.2115.
16. Nagoba B. Acidic Environment and Wound Healing: A Review. *Wounds* **2015**;27(1).
17. El-Kenawi A, *al. e*, Gillies RJ. Acidity promotes tumour progression by altering macrophage phenotype in prostate cancer. *Br J Cancer* **2019**;1121:556-66.
18. Andreev OA, Karabadzhak AG, Weerakkody D, Andreev GO, Engelman DM, Reshetnyak YK. pH (low) insertion peptide (pHLIP) inserts across a lipid bilayer as a helix and exits by a different path. *Proc Natl Acad Sci U S A* **2010**;107(9):4081-6 doi 10.1073/pnas.0914330107.
19. Tapmeier TT, Moshnikova A, Beech J, Allen D, Kinchesh P, Smart S, *et al.* The pH low insertion peptide pHLIP Variant 3 as a novel marker of acidic malignant lesions. *Proc Natl Acad Sci U S A* **2015**;112(31):9710-5 doi 10.1073/pnas.1509488112.
20. Narayanan T, Weerakkody D, Karabadzhak AG, Anderson M, Andreev OA, Reshetnyak YK. pHLIP Peptide Interaction with a Membrane Monitored by SAXS. *J Phys Chem B* **2016**;120(44):11484-91 doi 10.1021/acs.jpcc.6b06643.

REVIEWER COMMENTS

Reviewer #1 (Remarks to the Author):

The authors have answered my criticisms and I have no further comments.

Kevin Brindle

Reviewer #2 (Remarks to the Author):

In the revised manuscript, the authors have address part of my previous concerns, although not completely. In particular, for the in vivo relevance of T cell activation in affecting the lymph node pH, there are still no data supporting the conclusion (my original point #2). The authors should use antigen-specific T cells for adoptive transfer experiment, and examine in vivo pH with or without antigen stimulation.

Part of my original question #1 asked for the results from organs other lymph nodes, but this question was left unanswered.

For my original question #3, I understand the difficulty in making a definitive conclusion, but the authors should tune down their conclusions and impacts accordingly, to avoid over-interpretation of their results and conclusions.

Reviewer #3 (Remarks to the Author):

The authors have adequately addressed many of the issues raised in the original review. However, there are still a few remaining points that require attention:

Original comment: The authors show that acidity decreases T cell proliferation and cytokine production. But these are the main activities that are supposed to happen in the lymph node. Antigen presentation leads to activation and effector cell proliferation locally, before the cells leave the node. There are also critical paracrine effects of the cytokines that would be prevented by the authors' proposed mechanism. For example, helper T cells activated in the node produce cytokines that activate CD8 T cells and B cells in the same node. In this way, the lymph node is a factory of effector cells and antibodies that, if inhibited by acidic pH, would be expected to thwart systemic immunity. These issues are not discussed in the manuscript.

Author response: Our data are not suggesting that acidity decreases T cell proliferation. Indeed, Fig S11D showed that proliferation of activated CD4+ and CD8+ T cells was only modestly affected by acidity. To investigate this with more sensitive tools, we performed [³H]thymidine incorporation assays of naïve (CM), and peptide- or CD3-activated pMel, OT-I and OT-II cells. As shown in the figure below (provided for review), while there was more incorporation at pH 7.4, it was not substantial, consistent with the flow data of S11D.

Follow-up comment:

It is not clear what the authors mean by "modestly" or "substantial", but there appear to be statistically significance differences, and the authors state on lines 354-55: "In addition to its inhibitory effect on effector functions, acidosis also decreased the proliferation rate of T-cells."

Regarding the other effects of cytokines in the lymph node, the authors did not address this issue. I understand that regulation and function of the cytokine milieu in the lymph node is poorly understood and difficult to fully dissect in this study. However, the current text gives the impression that everything is controlled by pH, and no other mechanisms are needed. This could mislead readers. There must be other feedback systems preventing the T cells from mounting an

immune response within the lymph node. An obvious mechanism is the lack of appropriate target cells to kill there (i.e., when there is no intra-node infection).

Along these lines, the authors should discuss why, in the context of their theory, anti-inflammatory cytokines such as IL-4, IL-10 and IL-13 are also down-regulated at low pH (Figure S11B).

Additional comment: The authors should clarify which T cell "effector functions" they are addressing in the study. They are not presenting data on cytotoxic function or degranulation, but only production of cytokines (primarily IFgamma). The uninitiated reader might infer that low pH is preventing the T cells from actively killing LN bystander cells; but this is not really a danger, as there are other control mechanisms in place. The data show that pH is low in the LNs, and that T cells produce different cytokine levels at low pH. The authors should focus on this, and the potential ramifications of increased IFgamma in the node. Other speculation and language related to subsequent T cell effector function should be de-emphasized or removed.

REVIEWER COMMENTS

Reviewer #1 (Remarks to the Author):

The authors have answered my criticisms and I have no further comments.

Thank you

Reviewer #2 (Remarks to the Author):

1. In the revised manuscript, the authors have address part of my previous concerns, although not completely. In particular, for the *in vivo* relevance of T cell activation in affecting the lymph node pH, there are still no data supporting the conclusion (my original point #2). The authors should use antigen-specific T cells for adoptive transfer experiment, and examine *in vivo* pH with or without antigen stimulation.

In response to your previous point #2, we determined the absolute number of CD3+ T-cells in the LN and showed that, *even if they were all naïve*, there would be sufficient acid production to lower the pH to 6.2, the pH value we observe experimentally. The only difference between naïve and antigen-specific activated T cells is the time it would take to reach steady-state pH (10 hr and 2 hr, respectively). Thus, while we agree with the reviewer that antigen-specific activated T cells represent a very small population within the lymph node, we are not claiming that they are uniquely responsible for LN acidity. We explicitly state that even the reduced metabolism of naïve T cells is sufficient to acidify LNs, because T-cell metabolism continues to acidify the milieu until a set point of pH 6.2 is reached. Hence, the addition of a small subpopulation of activated antigen-specific T cells into the LN *in vivo* would not be expected to reduce pH any further. We would very likely see no difference in imaged pH.

In light of these considerations, we are unsure of the expected endpoint of the experiment requested by the reviewer.

2. Part of my original question #1 asked for the results from organs other lymph nodes, but this question was left unanswered.

We apologize. The second part of the original question #1 was "*they need to examine organs other than the lymph node to ascertain the specificity of the observed phenomena*". We unfortunately lost sight of that comment because we focused on the first part of question #1, which were the acute lymphodepletion experiments that were necessary to test the hypothesis that T-cells were contributing to the LN acidity.

With regard to other tissues, there is a long history by the Reshetnyak/Andreev group to investigate the biodistribution and pH-dependence of pHLIP accumulation. A few of those references have been provided out of the dozens that are germane. Demoin et al., *Bioconjugate Chem.*, presented a full tissue distribution of pHLIPvar3 (the variant

used in the current study) in Balb/c mice bearing 4T1 orthotopic tumors at times from 1-48 hr. At 24 hr (the time point used in the current study), tumors retained 19.6 ± 2.34 %ID/g; whereas only the clearance organs liver and kidney retained more than 10% at 13.6 ± 1.58 %ID/g and 13.3 ± 1.14 %ID/g, respectively. As the dye can affect the route of clearance, the addition of Cy5.5 (the dye used in the current study) moves the clearance organ more away from kidneys and more towards the liver. The following are ex vivo images of (clockwise from top left) liver, kidneys (2), gastrocnemius, and bilateral 4T1 mammary tumors (2) captured 24 hr post-injection of Cy5.5 coupled to WT pHLIP (green), Var3 (red), and Var7 (blue):

These studies did not explicitly investigate LNs. Thus, we performed BD studies of Tramp C2 tumor bearing mice. The figure below shows the BD of Cy 5.5 labeled Var3, including LNs. As above, the majority of signal was in the tumor and liver, and the normalized signal efficiency (per mm²) in Inguinal LNs (I-LN) was comparable to that of lung, pancreas, spleen, and kidneys.

3. For my original question #3, I understand the difficulty in making a definitive conclusion, but the authors should tune down their conclusions and impacts accordingly, to avoid over-interpretation of their results and conclusions.

Thank you. Reviewer #3 had a similar concern. We agree and have done so. We have re-worked much of the discussion to be less strident, and have added the following (new text in red):

“Whilst this low pH does not block the process of activation by antigen, it will suppress the production and release of many (but not all) cytokines, thereby possibly protecting the LN from premature and unwarranted release of inflammatory and anti-inflammatory cytokines. The complexity of these cytokines’ interactions within a LN are poorly understood and perhaps one function of this acid-induced inhibition of T-cells is just to simplify this milieu within the confined space of a LN”.

Reviewer #3 (Remarks to the Author):

Original comment: The authors show that acidity decreases T cell proliferation and cytokine production. But these are the main activities that are supposed to happen in the lymph node. Antigen presentation leads to activation and effector cell proliferation locally, before the cells leave the node. There are also critical paracrine effects of the cytokines that would be prevented by the authors’ proposed mechanism. For example, helper T cells activated in the node produce cytokines that activate CD8 T cells and B cells in the same node. In this way, the lymph node is a factory of effector cells and antibodies that, if inhibited by acidic pH, would be expected to thwart systemic immunity. These issues are not discussed in the manuscript.

Author response: Our data are not suggesting that acidity decreases T cell proliferation. Indeed, Fig S11D showed that proliferation of activated CD4+ and CD8+ T cells was only modestly affected by acidity. To investigate this with more sensitive tools, we performed [3H]thymidine incorporation assays of naïve (CM), and peptide- or CD3-activated pMel, OT-I and OT-II cells. As shown in the figure below (provided for review), while there was more incorporation at pH 7.4, it was not substantial, consistent with the flow data of S11D.

Follow-up comment:

It is not clear what the authors mean by “modestly” or “substantial”, but there appear to be statistically significance differences, and the authors state on lines 354-55: “In addition to its inhibitory effect on effector functions, acidosis also decreased the proliferation rate of T-cells.” The data presented in Figure 4F show that B6 cells stimulated with CD3 appear to have lower proliferation (less dilution) with CellTrace violet. To follow this up, we monitored the uptake of ³H Thymidine 24 and 48 hr after antigen-specific stimulation of Pmel, OT-1 and OT-II cells with CD3 or specific antigens: gp100₂₅₋₃₃ peptide, OVA_{SINFEKL} and OVA₃₂₃₋₃₃₉ peptide, respectively. At 24 hr, the only inhibition observed was with CD3 in Pmel cells. OT-I cells actually incorporated more TdR at pH 6.8. At 48 hours., stimulation of OT-I and OT-II cells with CD3 did

have a statistically significant inhibitory effect, but there was no difference in these cells when stimulated with specific antigen. The only antigen-specific inhibition by acid pH was observed in Pmel cells. The OT-I and OT-II results are consistent with those observed in B6 cells. Thus, while in some settings T cell proliferation was significantly decreased at low pH after stimulation with anti-CD3 antibody, these changes may not be biologically meaningful, as the results with anti-gen were either no effect or mixed. We have changed text to the following: “Although there was an inhibitory effect on cytokine secretion, acidosis did not consistently inhibit the proliferation rate of B6m Omel, OT-I, or OT-II T-cells stimulated with anti-CD3 antibodies (Fig 4F) or specific antigen (data provided with review)”.

1. Regarding the other effects of cytokines in the lymph node, the authors did not address this issue. I understand that regulation and function of the cytokine milieu in the lymph node is poorly understood and difficult to fully dissect in this study. However, the

current text gives the impression that everything is controlled by pH, and no other mechanisms are needed. This could mislead readers. There must be other feedback systems preventing the T cells from mounting an immune response within the lymph node. An obvious mechanism is the lack of appropriate target cells to kill there (i.e., when there is no intra-node infection).

Along these lines, the authors should discuss why, in the context of their theory, anti-inflammatory cytokines such as IL-4, IL-10 and IL-13 are also down-regulated at low pH (Figure S11B).

We initially speculated that activation of T cells at low pH led to a generalized suppression of cytokine production or secretion. When we performed the global analysis, we found that this was not true as some cytokines/chemokines were not affected at low pH. Further studies are needed to fully understand the role of pH on specific cytokine transcription and translation and the immuno-biological consequences. It could be, as the reviewer suggests, that these anti-inflammatory cytokines are more relevant in the LN, compared to IFN-g. Without more work on this subject it is difficult to know. To address this concern, we have throughout the text removed reference to “effector function”, and refer to it now as “cytokine production”.

2. Additional comment: The authors should clarify which T cell “effector functions” they are addressing in the study. They are not presenting data on cytotoxic function or degranulation, but only production of cytokines (primarily IFN γ). The uninitiated reader might infer that low pH is preventing the T cells from actively killing LN bystander cells; but this is not really a danger, as there are other control mechanisms in place. The data show that pH is low in the LNs, and that T cells produce different cytokine levels at low pH. The authors should focus on this, and the potential ramifications of increased IFN γ in the node. Other speculation and language related to subsequent T cell effector function should be de-emphasized or removed.

We have reworked the Discussion in response to this concern (as well as concern #3 or reviewer #2) and have made our conclusions less strident. We have also added (new text in red):

“Whilst this low pH does not block the process of activation by antigen, it will suppress the production and release of many (but not all) cytokines, thereby possibly protecting the LN from premature and unwarranted release of inflammatory and anti-inflammatory cytokines. The complexity of these cytokines’ interactions within a LN are poorly understood and perhaps one function of this acid-induced inhibition of T-cells is just to simplify this milieu within the confined space of a LN”.

REVIEWERS' COMMENTS:

Reviewer #2 (Remarks to the Author):

The authors have successfully addressed my concerns.

Reviewer #3 (Remarks to the Author):

The authors have adequately addressed my concerns.